**nature** COMMUNICATIONS

# Vascular and blood-brain barrier-related changes underlie stress responses and resilience in female mice and depression in human tissue

Laurence Dion-Albert [1], Alice Cadoret [1], Ellen Doney [1], Fernanda Neutzling Kaufmann [1], Katarzyna A. Dudek [1], Beatrice Daigle [1], Lyonna F. Parise [2], Flurin Cathomas [2], Nalia Samba [3], Natalie Hudson[4], Manon Lebel [1], Signature Consortium*, Matthew Campbell [4], Gustavo Turecki [5], Naguib Mechawar [5] & Caroline Menard [1✉]

Prevalence, symptoms, and treatment of depression suggest that major depressive disorders (MDD) present sex differences. Social stress-induced neurovascular pathology is associated with depressive symptoms in male mice; however, this association is unclear in females. Here, we report that chronic social and subchronic variable stress promotes blood-brain barrier (BBB) alterations in mood-related brain regions of female mice. Targeted disruption of the BBB in the female prefrontal cortex (PFC) induces anxiety- and depression-like behaviours. By comparing the endothelium cell-specific transcriptomic profiling of the mouse male and female PFC, we identify several pathways and genes involved in maladaptive stress responses and resilience to stress. Furthermore, we confirm that the BBB in the PFC of stressed female mice is leaky. Then, we identify circulating vascular biomarkers of chronic stress, such as soluble E-selectin. Similar changes in circulating soluble E-selectin, BBB gene expression and morphology can be found in blood serum and *postmortem* brain samples from women diagnosed with MDD. Altogether, we propose that BBB dysfunction plays an important role in modulating stress responses in female mice and possibly MDD.

[1] Department of Psychiatry and Neuroscience, Université Laval and CERVO Brain Research, Quebec City, QC, Canada. [2] Fishberg Department of Neuroscience and the Friedman Brain Institute, Icahn School of Medicine at Mount Sinai, New York, NY, USA. [3] Sorbonne Université, Paris, France. [4] Smurfit Institute of Genetics, Trinity College Dublin, Lincoln Place Gate, Dublin 2, Ireland. [5] Department of Psychiatry, McGill University and Douglas Mental Health University Institute, Montreal, QC, Canada. *A list of authors and their affiliations appears at the end of the paper. ✉email: caroline.menard@fmed.ulaval.ca

Major depressive disorder (MDD) is the most prevalent mood disorder and the leading cause of disability worldwide[1, 2]. In parallel, cardiovascular diseases are the main cause of years of life lost[3] highlighting the burden of these conditions. Prevalence of MDD is two to three-fold higher in patients suffering from cardiovascular diseases and it is associated with increased risk of morbidity and mortality[1, 4–6]. We previously reported that chronic social stress alters blood-brain barrier (BBB) integrity promoting depression-like behaviours in male mice[7, 8], indicating a direct link between neurovascular health and stress vulnerability. MDD is more frequent in women, who report higher levels of stress in daily life (American Psychological Association), the main environmental risk factor to develop depression[1]. Prevalence, symptoms, and treatment of MDD all point toward major sex differences with women more likely to experience comorbid anxiety, sadness, and social impairment[9–11]. However, it is yet unclear if chronic stress induces sex-specific neurovascular changes that could contribute to depression pathogenesis. MDD is an emerging nontraditional risk factor to develop cardiovascular diseases in women, particularly among young women who have higher rates of depression[12]; nevertheless, underlying mechanisms have yet to be determined[13].

Here we evaluated the effect of chronic social defeat stress (CSDS)[14] and chronic variable stress (CVS)[15], two mouse models of depression, on BBB-related gene expression, morphology and function. Chronic social stress is a prominent contributor to mood disorder prevalence and suicide attempts in victims of bullying[16]. Female rodents are more vulnerable to unpredictable stressors and develop anxiety- and depression-like behaviours after only 6 days (subchronic variable stress, SCVS) while males do not[15], allowing the identification of promising sex-specific targets. The BBB is formed by endothelial cells sealed by tight junction proteins, pericytes and astrocytes, and prevents potentially harmful signals in the blood from entering the brain[17]. The development of new and better targeted antidepressant drugs has been hampered by this barrier. Furthermore, its involvement in stress responses remains understudied, particularly when it comes to understanding sex and individual differences in BBB properties[18] as potentially associated to stress resilience vs vulnerability. To address these gaps, the present study uses complementary mouse models of depression and combined behavioural, molecular, morphological, and viral-mediatedloss-of-function experiments with endothelium-specific transcriptomic profiling to investigate vascular alterations underlying stress vulnerability vs resilience and gain mechanistic insights. Our results provide characterization and functional interrogation, in a region- and cell-specific manner, of the role played by the endothelium in chronic stress responses in female mice as well as in postmortem brain samples from women with MDD. We also identify circulating vascular potential biomarkers that could help better diagnose and inform treatment strategies for depressive disorders.

## Social stress induces regional neurovascular alterations in female mice and women with major depression disorder

In rodents, CSDS induces a depression-like phenotype mimicking human symptoms such as social avoidance, anhedonia, or anxiety in a subset of mice identified as stress-susceptible (SS)[14]. In the modified CSDS protocol, female C57BL/6 mice are exposed daily (10 min/day) to bouts of social defeat by a larger, physically aggressive CD-1 male mouse (Fig. 1a), after application of male CD-1 urine on the vagina, tail base and upper back of the female[19]. Male urine is essential to initiate aggression towards females and parallels the classic male/male 10-day CSDS

paradigm. A social interaction (SI) test is performed 24 h after the last exposure to social stress and defeated mice that do not display social avoidance are considered resilient (RES) (Fig. 1b and Supplementary Fig. 1a, b, $p < 0.0001$). First, we aimed to identify individual differences in the neurovasculature potentially underlying stress responses in SS vs RES females. Transcriptional profiling of genes associated with vascular integrity, permeability, angiogenesis, tight junctions, and BBB formation was performed in the nucleus accumbens (NAc) of unstressed control (CTRL), SS and RES mice after 10-day CSDS (Fig. 1c; quantitative PCR primers are listed in Supplementary Table 1). Indeed, the NAc plays crucial roles in reward processing, stress responses and mood disorders, including MDD[20], and we reported in a previous study[7] that the vasculature of this brain region is altered in SS males. In contrast to their male counterparts, we did not observe significant changes for BBB-related gene expression in the NAc of females (Fig. 1c and Supplementary Fig. 1c), including for the tight junction Claudin-5 (Cldn5) (Fig. 1d), a gatekeeper of BBB permeability associated with depression-like behaviours in male mice[7]. MDD clinical features are characterized by sexual disparities as well as sex-specific regional transcriptional signatures[9], thus, we explored if the neurovasculature could be affected in another mood-related hub, the prefrontal cortex (PFC). This brain region is involved in social behaviours, executive function and decision making[20–23]. Major alterations were observed in the PFC of stressed females (Fig. 1e and Supplementary Fig. 1d) including a ~50% loss of Cldn5 for SS mice (Fig. 1f, $p = 0.0042$). This vascular change may be sex-specific given that no difference was noted in the PFC of stressed males[7]. Since various models of CSDS have been developed for females in recent years, we confirmed that vascular alterations are also present in a different social defeat paradigm. In this protocol, male aggression towards females is initiated through chemogenetic activation of the ventrolateral subdivision of the ventromedial hypothalamus (Supplementary Fig. 2a)[24]. Behavioural phenotyping of female mice exposed to this 10-day social defeat paradigm using the SI test revealed two subpopulations of stressed animals, SS and RES mice (Supplementary Fig. 2b, $p < 0.0001$), as observed with the other paradigm described above. Stress susceptibility was again associated with a loss of Cldn5 in mood-related brain regions, including the PFC (Supplementary Fig. 2c, $p = 0.0122$), with no difference for RES females when compared to unstressed CTRL, confirming that vulnerability to chronic social stress is linked to alterations in the female brain vasculature.

Next, we confirmed that this stress-induced region-specific change observed in females is also present at the protein level using double immunostaining with the endothelial cell marker cluster of differentiation 31 (CD31) (Fig. 1g). Following 10-day CSDS, loss of Cldn5 in the female PFC was present only in SS and not RES mice when compared to unstressed CTRL (Fig. 1h, $p = 0.0005$) and significantly correlated with social interactions (Fig. 1h and Supplementary Fig. 2d–g). No difference was observed in the NAc of stressed females (Supplementary Fig. 2h, i). Finally, we assessed the translational value of our mouse findings by evaluating CLDN5 expression in postmortem ventromedial PFC tissue from individuals with MDD who died by suicide. We confirmed ~50% loss of CLDN5 in women with MDD when compared to matched nonpsychiatric controls (left, $p = 0.0068$) along with alterations in vessel morphology (right, $p = 0.0288$) (Fig. 1i, Supplementary Table 2 for qPCR primers and Supplementary Table 3 for demographics). Conversely, no significant difference was observed for men with MDD in this brain region (Fig. 1i, $p = 0.5387$). Together these findings suggest that chronic stress induces region-specific BBB alterations that may be involved in sex differences in MDD symptomatology.

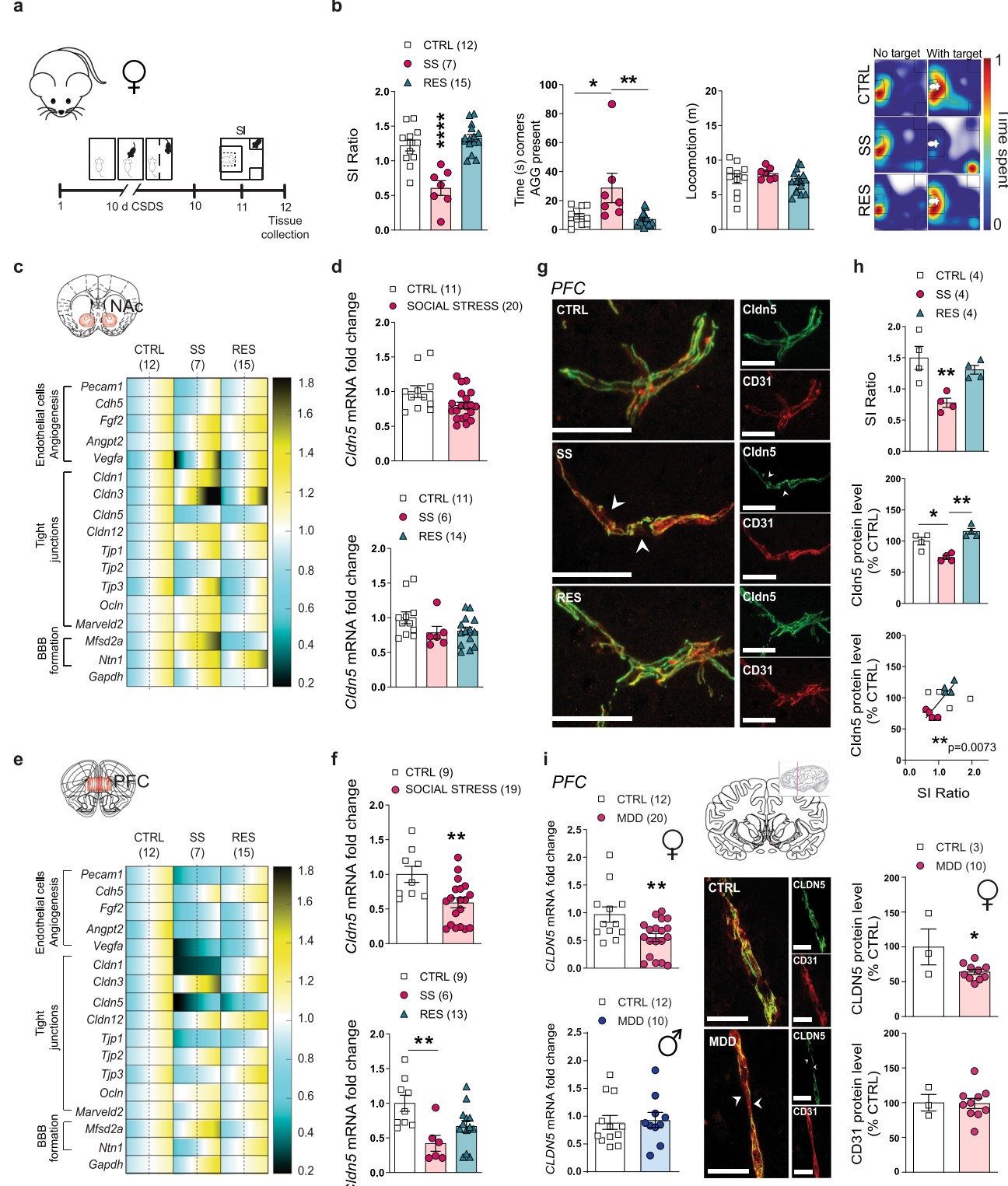

## Region-specific vascular changes promoted by subchronic variable stress in line with anxiety- and depression-like behaviours

Female rodents are more vulnerable to unpredictable stress with 6 consecutive days of variable stressors being sufficient to induce anxiety- and depression-like behaviours in females but not males[15], allowing for an exploration of the mechanisms underlying sex-specific stress responses. Female mice were exposed to a series of three different stressors namely foot shocks, tail

suspension or restraint stress (1/day), repeated twice and then subjected to a battery of behavioural tests (Fig. 2a). Exposure to this subchronic variable stress (SCVS) paradigm induced social avoidance (Fig. 2b and Supplementary Fig. 3a, $p = 0.0323$ for time spent in the interaction zone when the aggressor is present), anxiety in the elevated plus maze (Fig. 2c, Supplementary Fig. 3b, $p = 0.026$ for time spent in closed arms) and anhedonia in the sucrose preference test (Fig. 2d, $p = 0.0004$, $p = 0.0007$ and $p = 0.0042$ for 24 h, 48 h and in average, respectively). Stress-

**Fig. 1 Chronic social stress induces region-dependent neurovascular changes in female mice. a** Experimental timeline of 10-day chronic social defeat stress (CSDS), social interaction (SI) and tissue collection of nucleus accumbens (NAc) and prefrontal cortex (PFC). **b** Individual SI values (left, ****$p < 0.0001$), time (s) in corners with aggressor (AGG) present (middle, **$p = 0.0038$) and representative heatmaps of normalized time spent in the arena during SI test (right). **c** Quantitative PCR revealed changes in the NAc of stress susceptible (SS) and resilient (RES) mice when compared to unstressed controls (CTRL) of gene expression related to endothelial cells, angiogenesis, tight junctions, and blood-brain barrier (BBB) formation, (**d**) but *Cldn5* levels remained unchanged ($p = 0.1015$ for CTRL vs SOCIAL STRESS; $p = 0.1262$ for CTRL vs SS vs RES). The range of color indicates individual differences within a group; average represented by the dashed line. **e** Quantitative PCR in the PFC revealed region-specific changes in SS and RES mice when compared to unstressed CTRL, and *Cldn5* mRNA (**$p = 0.0042$ for CTRL vs SOCIAL STRESS; **$p = 0.0055$ for CTRL vs SS vs RES) (**f**) and protein levels (**g, h**) were lower in the PFC of SS mice (***$p = 0.0005$) and correlated with social avoidance (**$p = 0.0073$). Scale bars, 20 μm. **i** *CLDN5* mRNA fold change was significantly lower in the PFC of women with major depressive disorder (MDD) (left, **$p = 0.0068$) along with morphological vascular alterations (right, *$p = 0.0288$). No significant difference was observed for men (left, $p = 0.5387$). Scale bars, 20 μm. Data represent mean ± s.e.m; the number of animals or subjects (*n*) is indicated on graphs. Correlations were evaluated with Pearson's correlation coefficient; 2-group comparisons were evaluated with unpaired t-tests and one-way ANOVA followed by Bonferroni's multiple comparison test for other graphs. *$p < 0.05$; **$p < 0.01$; ****$p < 0.0001$. Source data are provided as a Source Data file.

induced alterations in these behavioural domains were correlated with each other particularly for anhedonia (Supplementary Fig. 3c). Along with the establishment of anxiety- and depression-like behaviours, 6-d of SCVS instigates changes in the neuro-vasculature of the female brain. Transcriptomic studies of BBB associated genes revealed ~30% loss for *Cldn5* in the NAc of stressed mice (Fig. 2e, f, $p = 0.0506$) with greater downregulation in the PFC (~40%) when compared to unstressed controls (Fig. 2g, h, $p = 0.0125$). No significant difference was observed for the endothelial cell marker CD31 (*Pecam1*, Fig. 2e–h) or other tight junctions (Supplementary Fig. 3f, g). We again evaluated the translational value of our mouse findings in *postmortem* brain samples from MDD women and men and observed a loss of *CLDN5* at the mRNA level (Fig. 2i, $p = 0.0275$ and $p = 0.0253$, respectively). However, the morphology of the vessels was not significantly different in the female brain (Fig. 2j, $p = 0.0557$), suggesting that this brain region could be less vulnerable than the ventromedial PFC.

### Downregulation of tight junction *Claudin-5* expression in the prefrontal cortex promotes anxiety-like and depression-like behaviours, including social avoidance, in females

To confirm that loss of Cldn5 in the female PFC plays a causal role in the establishment of anxiety- and depression-like behaviours we used an adeno-associated virus (AAV)-mediated approach to conduct functional studies (Fig. 3a). We chose this approach since *Cldn5*-deficient mice die within 10 h of birth[25] and stress paradigms were performed in adult mice. It also allows downregulation of *Cldn5* expression in a brain region and cell-specific manner[7], with this tight junction only expressed in endothelial cells[26], and functional BBB deficits with leakage of circulating dyes or proteins into the brain[7, 27]. We first confirmed the efficiency of Cldn5 conditional knockdown at mRNA ($p = 0.0021$) and protein ($p = 0.0403$) level following injection of an AAV2/9 serotype expressing a doxycycline-inducible*Cldn5*-targeting shRNA (AAV-shRNA-*Cldn5*)[27] in this brain region (Fig. 3b). Next, another cohort of female mice injected with either an AAV-shRNA-*Cldn5* or an AAV-shRNA (control) virus[27] were subjected to a battery of behavioural tests[7] (Fig. 3a). Half the mice were exposed to a micro-defeat prior to behavioural testing. This acute stress does not induce anxiety- or depression-like behaviours in naive mice but is commonly used to reveal a pro-susceptible phenotype[7]. Viral-mediated downregulation of *Cldn5* expression in the female PFC decreased time spent in the open arms of the elevated plus maze (Fig. 3c and Supplementary Fig. 4a, b), time spent grooming in the splash test (Fig. 3d), sucrose consumption (Fig. 3e and Supplementary Fig. 4c) and increased immobility time in the forced swim test (Fig. 3f). On the other hand, social interactions were reduced in both virus-injected groups following a micro-defeat (Fig. 3g). A significant

virus effect was observed for most behavioural tests with no difference between unstressed vs stressed AAV-shRNA-*Cldn5*-injected animals, indicating that artificial opening of the BBB in the PFC is sufficient to induce anxiety- and depression-like behaviours in female mice without prior acute stress exposure (Fig. 3h, $p = 0.0129$ for elevated plus maze open arms, $p = 0.0084$ for splash test, $p < 0.0001$ for sucrose preference, $p = 0.0453$ for forced swim test). As shown in Fig. 3i and Supplementary Fig. 4d, viral-mediated loss of *Cldn5* in the female PFC affects multiple behavioural domains revealing a central role of the BBB in this brain region in mediating stress responses. Finally, we explored further the impact of *Cldn5* downregulation in the PFC on social interactions by injecting a separate cohort of female mice with AAV-shRNA-*Cldn5* or control AAVs and exposing them to CD1 mice of both sexes in a SI test (Fig. 3j). Reduction of *Cldn5* expression in the female PFC induced social avoidance when mice were given the opportunity to interact with another female ($p = 0.0123$) but not with a male ($p = 0.3016$) (Fig. 3k and Supplementary Fig. 5a, b). Social interactions were significantly correlated between sexes with mice injected with the AAV-shRNA-*Cldn5* virus interacting less than the AAV-shRNA controls (Fig. 3l and Supplementary Fig. 5c). Importantly, loss of *Cldn5* expression is significantly correlated with social avoidance observed towards female mice (Fig. 3l). Overall, these results suggest that loss of BBB integrity in the female PFC could play a key role in the pathogenesis of maladaptive stress responses and mood disorders.

### Female endothelium transcriptomic profiles associated with resilience vs the establishment of depression-like behaviours following chronic stress exposure

To gain mechanistic insights on the effects of chronic social and variable stress on BBB biology and properties we performed transcriptome-wide gene expression profiling of female PFC endothelial cells. Female mice were subjected to 10-d CSDS, behavioural phenotype was defined using the SI test and PFC punches were collected 24 h later and immediately processed through magnetic-activated cell sorting (MACS) which exploits immunomagnetic microbeads to quickly and gently separate cell types[7, 28–30] (Fig. 4a and Supplementary Fig. 6a, b). Enrichment of endothelial cells and genes specific to this cell population was confirmed by fluorescence-activated cell sorting and qPCR, respectively (Supplementary Fig. 6c)[8]. RNA was extracted from female PFC endothelial cells of unstressed CTRL, SS and RES mice (Fig. 4b, $p < 0.0001$) and transcriptome profiles established with the mouse Clariom S assay, which allows measurement of gene expression from > 22,000 well-annotated genes[8, 28]. Hierarchical clustering of endothelium gene expression variations revealed low overlap between SS and RES groups with fold changes often going in the opposite direction when compared to

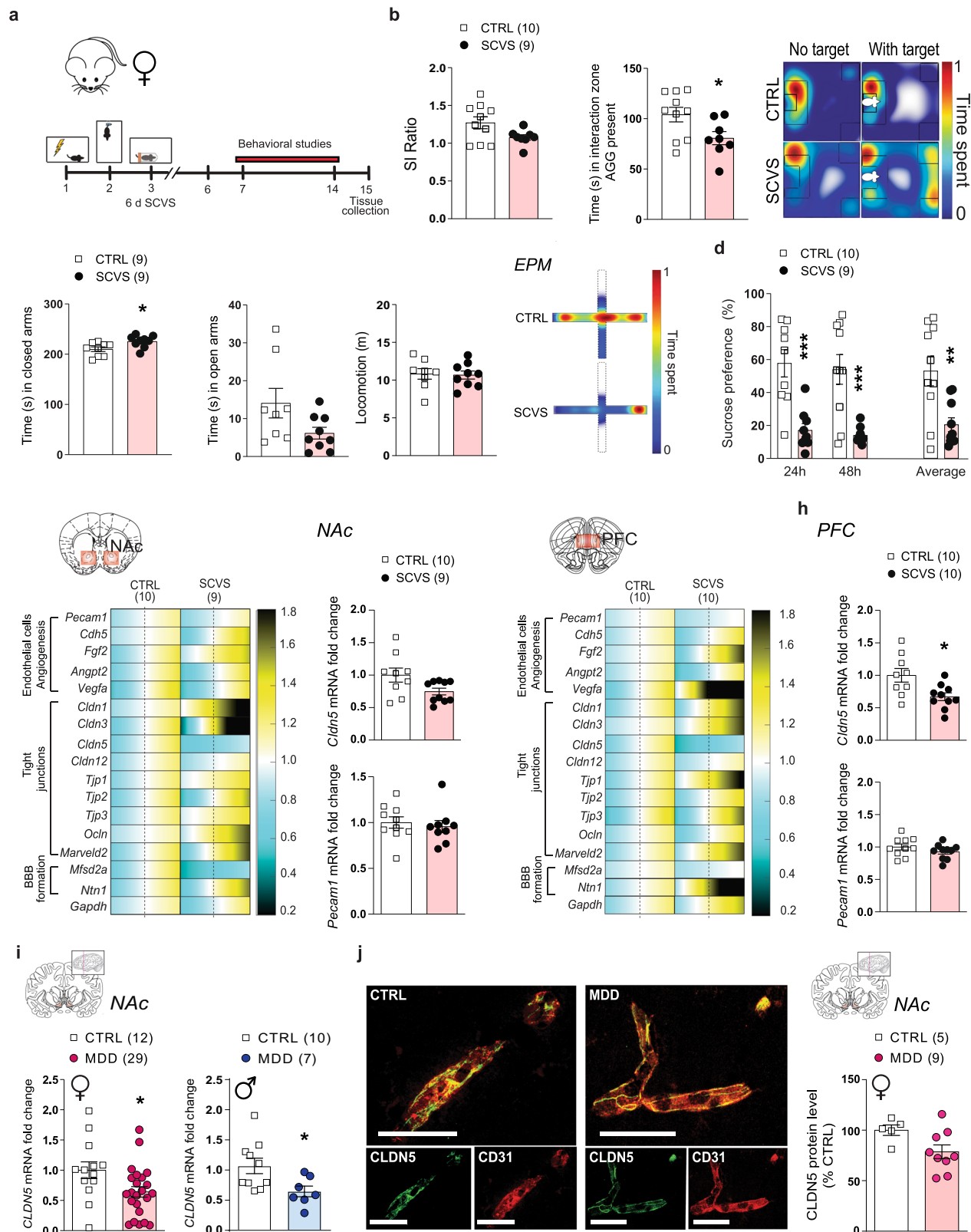

unstressed CTRL (Fig. 4c). These findings confirm that chronic social stress induces BBB adaptations underlying individual differences in stress responses. Analysis of biological pathways differentially regulated between PFC endothelial cells of SS vs CTRL female mice revealed increased expression of genes associated with oxidative damage while omega-3/omega-6 fatty acid synthesis was linked with resilience (Fig. 4d). Glial cell line-derived

(Gdnf)-Ret signalling (Fig. 4d, $p = 0.0141$) and genes linked to angiogenesis, cell migration and polarization (*Cytl1, Robo2, Fat4, Dock4*), BBB transport (*Cav2, Slc25a20*), permeability and inflammation (*Lcn2, Pla2g7, Vegfa, Ccl19, Ccl21a*) were differentially regulated in the female SS PFC when compared to RES animals (Fig. 4e). Gdnf, a neurotrophic factor, can alter *Cldn5* expression[31] while a proinflammatory environment promotes

**Fig. 2 Six-day chronic variable stress induces behavioural and region-dependent neurovascular changes in female mice. a** Experimental timeline of 6-d chronic variable stress (SCVS) and behavioural studies. **b** SCVS induces slight social interaction deficits ($p = 0.0548$) and decreased time in the interaction zone in stressed female mice when the social target (aggressor, AGG) is present (*$p = 0.0323$), in the social interaction test. 6-d of CVS is enough to induce significant anxiety- and depression-like behaviours in the elevated plus maze (**c**) (*$p = 0.026$) and sucrose preference (**d**) tests (***$p = 0.0004$ for 24 h; ***$p = 0.0007$ for 48 h; **$p = 0.0048$ for average). **e** Quantitative PCR of genes related to endothelial cell biology, angiogenesis, tight junctions and blood-brain barrier (BBB) formation reveals significant changes in the nucleus accumbens (NAc) of stressed female mice (**f**), and downregulation of *Cldn5* mRNA levels. **g** Quantitative PCR of those genes in the prefrontal cortex (PFC) reveals region-specific neurovascular changes in stressed female mice vs controls (CTRL) (**h**) and significant downregulation of *Cldn5* mRNA expression (*$p = 0.0125$). **i** *CLDN5* mRNA fold change was significantly lower in the NAc of men and women with major depressive disorder (MDD) (*$p = 0.0275$), but (**j**) immunostaining revealed no significant difference in CLDN5 expression in the brain of women with MDD. Scale bars, 20 μm. Data represent mean ± s.e.m; the number of animals or subjects (*n*) is indicated on graphs. 2-group comparisons were evaluated with unpaired t-tests. *$p < 0.05$; **$p < 0.01$; ***$p < 0.001$. Source data are provided as a Source Data file.

BBB hyperpermeability through loss of structural integrity and tight junction disassembly[7, 32].

We next sought to determine if exposure to a different type of chronic stressor would lead to a common or divergent endothelium transcriptomic profile in animals characterized by a depressive phenotype. Female mice were subjected to the 6-d SCVS paradigm (Fig. 5a) then PFC punches were collected 24 h after the last stressor and endothelial cells were isolated using MACS. RNA was extracted and endothelium transcriptomic profiles were produced using the mouse Clariom S assay as described above. As expected, stressed animals (SCVS) clustered together vs unstressed CTRL (Fig. 5b). However, when hierarchical clustering was performed on samples from mice exposed to either chronic social or variable stress, we noted similarities between the SS and SCVS groups, both characterized by depression-like behaviours, when compared to the RES animals (Fig. 5c). Alignment of PFC gene expression changes normalized on RES female mice revealed an overlap in the SS/SCVS directionality and amplitude (Fig. 5c), suggesting that different types of stressors can lead to similar BBB-related gene alterations underlying stress vulnerability in females. The endothelium transcriptomic changes induced by chronic stress revealed a poor overlap between males and females in the PFC (Supplementary Fig. 6d–g) or vulnerable brain regions (NAc in males and PFC in females) despite exposure to the same stressor (Supplementary Fig. 7a–c). To perform these analyses, a cohort of C57BL/6 male mice was subjected to 10-d CSDS, behavioural phenotype defined using the SI test (Supplementary Fig. 6d) and PFC punches collected 24 h later and immediately processed through MACS to establish transcriptomic profiles with the mouse Clariom S assay. Male NAc endothelium-related gene lists were obtained from our recent study[8]. In fact, major sex differences are also present in the PFC endothelium of unstressed controls with over 1700 differentially expressed genes (Supplementary Fig. 7d). Moreover, by comparing *Cldn5* expression at baseline we noted ~25% more in the PFC ($p = 0.0044$) and NAc ($p = 0.0034$) of unstressed females when compared to their male counterparts (Supplementary Fig. 7e). Differential expression of this tight junction protein is correlated between these two mood-related brain regions (Supplementary Fig. 7e, $p = 0.0089$) possibly in line with the individual differences observed following chronic stress exposure. In the female PFC, transforming growth factor beta (TGFβ) signalling was associated with SCVS (Fig. 5d, $p < 0.0001$) along with a decreased expression of genes involved in the Wnt signalling pathway (*Ctnnb1, Axin2*) or maintenance of BBB integrity (*Tjp1, Yap1*) and increased expression of inflammatory mediators (*Il18, Ccl19, Ccr5, Hsp90b1*) (Fig. 5e). We explored publicly available human MDD databases to confirm the translational value of our mouse sequencing-related findings[9]. Similar to mice, endothelium gene expression is altered according to sex in mood-associated brain regions of women with MDD (*CAV2, CCR5, CTNNB1, IL18, TJP1, TJP2*) vs men (*CLDN12, HSP90B1, VEGFA, YAP1*)

with poor overlap for both (*DOCK4, PLA2G7*). TGFβ and Wnt signalling are both directly associated with angiogenesis and maintenance of BBB integrity through expression of tight junction proteins, including Cldn5[33, 34], suggesting that exposure to chronic stress could lead to BBB hyperpermeability in the female PFC.

## Chronic stress induces BBB leakiness in female mice and changes in blood-based vascular biomarkers in the stress-susceptible mice also observed in women with major depression disorder

Considering that both 10-d CSDS and 6-d SCVS paradigm induced loss of *Cldn5* expression, blood vessel morphology alterations and proinflammatory transcriptomic changes in the female PFC, BBB permeability in this brain region was evaluated using peripheral injection of a fluorescent-tagged dextran. Peripheral injection of dyes such as a fluorophore-tagged 10 kDa dextran is a standard protocol to detect BBB disruption and hyperpermeability[33, 35]. First, we confirmed that retro-orbital injection of a 10 kDa lysine-fixable dextran conjugated with an AlexaFluor488 (AF488-dextran) can fill blood vessels of the PFC but cannot readily cross the BBB in the absence of neurovascular damage (Supplementary Fig. 8a). Next, female mice were subjected to 10 days of CSDS, susceptibility and resilience were established with a SI test ($p = 0.0273$), then 24 h later mice were injected retro-orbitally with the AF488-dextran 30 min prior perfusion to flush remaining circulating dye and brain fixation (Fig. 6a and Supplementary Fig. 8b). No dye infiltration was observed for the unstressed CTRL confirming intact BBB integrity in the PFC of female mice in the absence of stress exposure (Fig. 6a, b). Conversely, we found leaky vessels in both stressed groups with greater permeability for SS animals. The average fluorescence intensity of the 10kDa-AF488 dextran in leaky areas was higher in stressed mice when compared to unstressed controls ($p = 0.0144$); however, the leakiness was more frequent ($p = 0.0005$) in the PFC blood vessels of SS females (Fig. 6a, b, and Supplementary Fig. 8c, d) in line with the finding of stress-induced loss of Cldn5 (Fig. 1e–h). We previously reported that chronic social stress reduces BBB integrity in the NAc of SS, but not RES, male mice along with no alterations observed in the PFC for either group[7]. Our findings highlight the importance of considering sex differences when investigating the role of the neurovasculature in stress responses and mood disorders.

One of the few studies investigating basal sex differences in human BBB permeability in healthy volunteers reported a difference in cerebrospinal fluid/serum albumin ratio[36]. Furthermore, identification of MDD-related biomarkers is greatly needed to help guide clinical diagnosis. Thus, we explored the potential of various vascular biomarkers to determine stress susceptibility vs resilience in our mouse models of depression. We took advantage of a commercially available MILLIPLEX MAP Mouse Cardiovascular Disease Magnetic Bead Panel including 7 analytes related

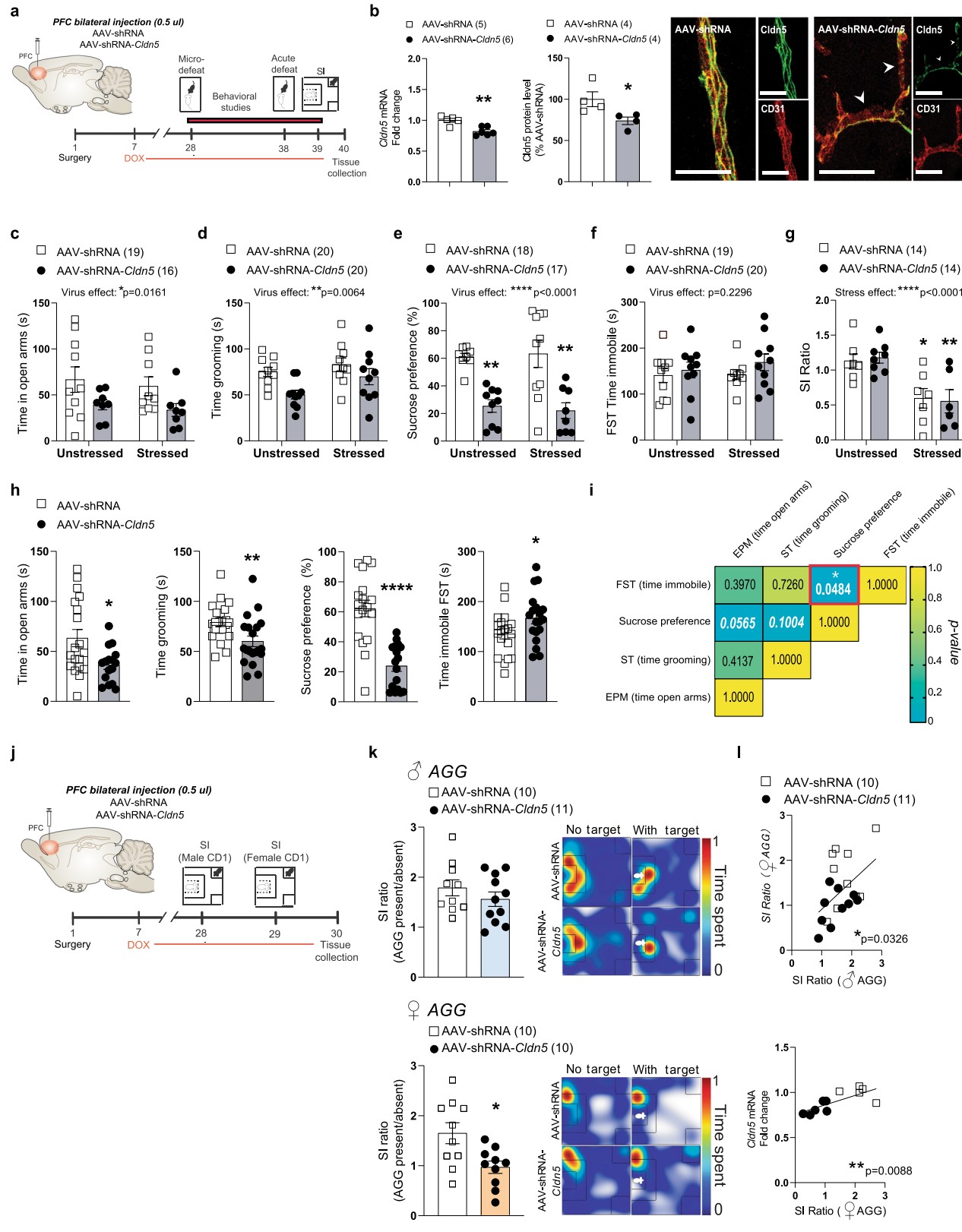

to vascular health. Female mice underwent the 10-d CSDS or 6-d SCVS paradigm and blood was collected 3 days before the first exposure to a stressor and 24 h after the last stress (Fig. 6c, d and Supplementary Fig. 9a–c). Social and variable stress were associated with significant changes in soluble adhesion molecule sE-selectin (CSDS: $p = 0.0267$), soluble plasminogen activator inhibitor-1 (sPAI-1, CSDS: $p = 0.0004$, SCVS: $p = 0.0010$), pro-

matrix metallopeptidase 9 (pro-MMP9, CSDS: $p = 0.0245$) and soluble thrombomodulin (sThrombo, CSDS: $p = 0.0360$) in females as measured with a Milliplex cardiovascular panel (Fig. 6c–e and Supplementary Fig. 9d–i). No significant difference was observed between SS and RES animals except for sE-selectin which was increased in the blood serum of SS female mice only after 10-d CSDS ($p = 0.0131$) and negatively correlated with SI

**Fig. 3 Conditional knockdown of *Cldn5* expression in the PFC is sufficient to induce anxiety- and depressive-like behaviours. a** Experimental timeline of prefrontal cortex (PFC) bilateral injection of AAV-shRNA of AAV-shRNA-*Cldn5* viruses and behavioural studies. **b** *Cldn5* mRNA (**$p = 0.0021$) and protein (*$p = 0.0403$) levels are reduced following AAV-shRNA-*Cldn5* injection in the PFC of female mice compared to that in AAV-shRNA-injected mice, following doxycycline (Dox) treatment. Scale bars, 20μm. Subthreshold microdefeat (stressed mice) did not have a significant effect on anxiety- and depressive-like behaviours in the elevated plus maze (**c**), splash test (**d**), sucrose preference test at 48 h (**$p = 0.0064$ unstressed AAV-shRNA vs AAV-shRNA-*Cldn5*; **$p = 0.0034$ stressed AAV-shRNA vs AAV-shRNA-*Cldn5*) (**e**), forced swim test (f, FST) and for social interactions (*$p = 0.0317$ unstressed AAV-shRNA vs stressed AAV-shRNA; **$p = 0.0089$ unstressed AAV-shRNA-*Cldn5* vs stressed AAV-shRNA-*Cldn5*) (**g**) tests. **h** However, main virus effects are observed when comparing all AAV-shRNA vs AAV-shRNA-*Cldn5* animals (*$p = 0.0129$ for elevated plus maze; **$p = 0.0084$ for splash test; ****$p < 0.0001$ for sucrose preference at 24 h; *$p = 0.0453$ for forced swim test). **i** Intraindividual correlation of different behavioural data points reveals trends between anxiety- and depressive-like behaviours. *P* values in the boxes refer to the strength of the correlation between behaviours. **j** Experimental timeline of PFC bilateral injection of AAV-shRNA or AAV-shRNA-*Cldn5* viruses and social interaction (SI) tests. **k** No significant difference in SI ratio was observed when a male social target (aggressor, AGG) was present, but a significant difference was found when a female AGG was present (*$p = 0.0123$). **l** SI ratios with male and female AGG values are significantly correlated to each other, and with *Cldn5* mRNA levels. Data represent mean ± s.e.m; the number of animals or subjects (*n*) is indicated on graphs. Two-way ANOVA followed by Bonferroni's multiple comparison test for behaviours, Correlations were evaluated with Pearson's correlation coefficient; 2-group comparisons were evaluated with unpaired t-tests. *$p < 0.05$; **$p < 0.01$; ****$p < 0.0001$. Source data are provided as a Source Data file.

ratio ($p = 0.0488$) (Fig. 6e, f). This vascular biomarker was also elevated after 6-d SCVS without reaching significance (Fig. 6g, $p = 0.1829$). Changes in circulating vascular biomarkers are different in males after exposure to the 10-day CSDS paradigm (Supplementary Fig. 10a) with stressed-induced variations observed for soluble intercellular adhesion molecule-1 (sICAM-1, $p = 0.0139$) and pecam-1 (sPecam-1, $p = 0.0296$) but not for sE-selectin ($p = 0.7246$) (Fig. 6h and Supplementary Fig. 10b–d). In fact, most vascular-related soluble molecules measured were characterized by sex differences at baseline (sE-selectin, $p < 0.0001$) (Fig. 6i and Supplementary Fig. 10e). The translational value of sE-selectin as a potential biomarker of mood disorder was confirmed on blood samples obtained from individuals with MDD (Fig. 6j, women: $p = 0.0494$, men: $p = 0.7859$). Like in mice, sE-selectin is lower at baseline in women of the healthy control group when compared to their male counterparts (Fig. 6k, $p = 0.0499$) supporting that vascular sex differences could underlie MDD.

## Discussion

Only a handful of studies have explored BBB sex differences, most indirectly and in vitro[37] but, to our knowledge, none did so in the context of chronic stress in mice or MDD. Overall, our findings indicate that chronic social and subchronic variable stressors alter BBB integrity in the mouse female brain through loss of the tight junction protein Cldn5 in the PFC and, to some extent, other mood-related brain regions such as the NAc. Importantly, these vascular alterations are also present in *postmortem* human brain samples from women with MDD. In mice, viral-mediated downregulation of *Cldn5* in the PFC is sufficient to promote anxiety- and depression-like behaviours including social avoidance, anhedonia, and helplessness supporting a causal role in the establishment of maladaptive stress responses and possibly, mood disorders. We did not observe stress-induced BBB dysfunction in the male PFC[7] indicating that chronic stress and depression affect the neurovasculature in a sex-specific manner. Different stress paradigms elicit specific anxiety- and depression-like behaviours according to sex, each recapitulating certain aspects of the symptoms and molecular features of MDD[38].

A potential limitation of the female CSDS model we have used here is exposure to a male stressor (i.e., antagonistic social confrontations by a larger male aggressor). In male C57BL/6, the CSDS paradigm has been shown to have relevant etiological, predictive and face validity[14]. However, additional studies are warranted to confirm that this model is also etiologically relevant for female mice. Furthermore, SI tests were conducted using a male social target and we cannot exclude a possible confounding

effect of intersex social interaction on the SI ratio values. Performing these tests with a female target may result in a different stratification of SS and RES mice, as we report for our virus-injected cohorts. We chose to use male mice as social targets in other contexts because social stress was performed by male aggressors[19]. Physical injuries are always a concern when running the CSDS paradigm either in males or females. Indeed, sickness behaviours could account for behavioral changes through stimulation of the immune system and entry of inflammatory mediators into the brain via altered BBB permeability. Therefore, physical examinations of animals were performed and no difference in the number of wounds was observed (Supplementary Figs. 1a and 8b) suggesting that susceptible vs resilient behavioral phenotypes were not due to physical injury. Finally, stress resilience is a fluid concept, and it is still debated whether or not it can be defined as a trait[39]. The complex neurobiological interactions underlying SS and RES phenotypes are not fully understood and are highly context-dependent. Coping strategies (i.e., social interactions) may be considered adaptive in some circumstances and maladaptive in others, and such mechanisms have been poorly investigated in females[40]. Thus, further studies are needed to better understand individual differences in stress-induced resilience vs vulnerability and how these behavioural responses and underlying mechanisms relate to human mood disorders.

Only females are susceptible to 6-d SCVS; however, both sexes display depression-like behaviours after weeks of stress exposure[9, 15] or if behavioural testing is performed 30 days after the 6-d SCVS paradigm[41]. Discrepancies in these behaviours could explain the sex-specific regional vascular effects observed and, possibly, sex differences associated with MDD. Magnetic resonance imaging (MRI) studies also support this idea, with unmedicated women with MDD showing decreased grey matter volume in limbic regions, including the left ventral PFC, while this reduction is observed in striatal regions for men with MDD[42]. Our study is also in line with recent clinical observations reporting region-specific BBB disruption in psychiatric disorders[43, 44], although sex differences were not addressed.

Our work not only highlights fundamental sex differences in stress-induced neurovascular responses but also provides mechanistic insights by identifying key pathways and genes involved. Bulk tissue sequencing studies show a major rearrangement of transcriptional patterns in mood-related brain regions in MDD with low overlap between men vs women with MDD (~10%)[9, 11]. These marked sex differences are also observed in mouse models of depression[9, 15]. Neuronal contribution is undeniable, notably via changes in neurotransmitter systems[9].

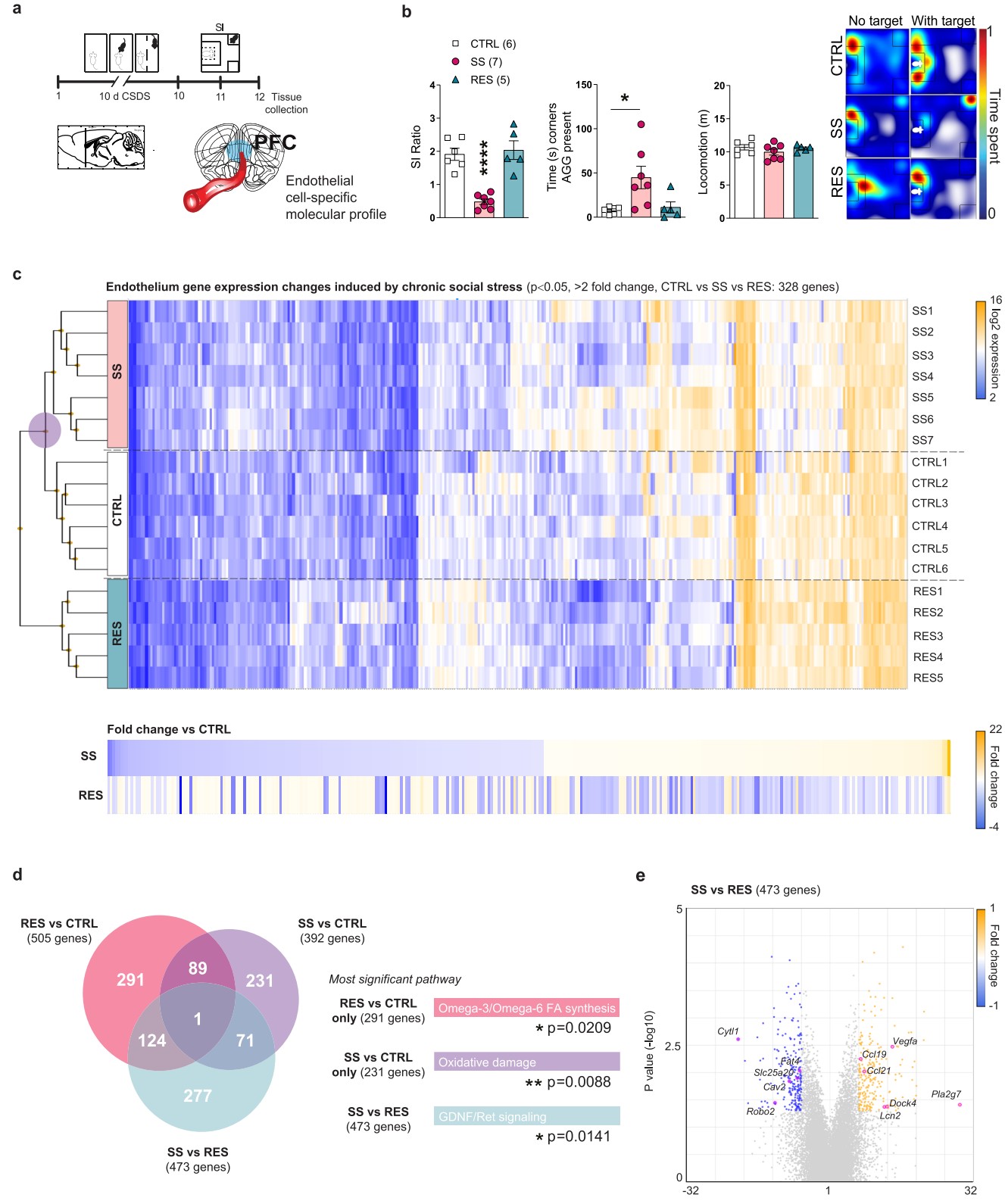

**c** Endothelium gene expression changes induced by chronic social stress (p<0.05, >2 fold change, CTRL vs SS vs RES: 328 genes)

Nevertheless, a significant enrichment for endothelium-related genes is also present[9] but had never been explored. Bulk RNA sequencing revealed a similar enrichment for genes related to this cell population in schizophrenia[45], reinforcing the involvement of the neurovasculature in psychiatric disorders. Although the resilience phenotype in female mice is not as clearly defined as it is for their male counterparts when analysing behaviours[19, 24],

our endothelium transcriptomic profiling revealed a distinct resilience-associated pattern when compared to animals displaying anxiety- or depression-like behaviours induced by either chronic social or subchronic variable stress. We found secreted enzyme phospholipase A2 Group VII (*Pla2g7*) as a gene significantly upregulated in SS vs RES mice, and *Pla2g7* methylation is associated with increased risk of coronary heart disease

**Fig. 4 Susceptibility vs resilience to chronic social stress is associated with specific endothelium transcriptome-wide changes in the female PFC.**
**a** Experimental timeline of 10-d chronic social defeat stress (CSDS) and (**b**) phenotype in the social interaction (SI, ****$p < 0.0001$; *$p = 0.0178$ for middle panel) test of the mice used to establish prefrontal cortex (PFC) endothelial gene profiles. **c** Hierarchical clustering heatmap of unstressed controls (CTRL), stress-susceptible (SS) and resilient (RES) mice, reveals that the RES group is the most distinct (purple circle, significance was set at ±2-fold change and $p < 0.05$). In fact, normalization of gene expression changes on the CTRL group shows that in stressed mice a majority of genes are regulated in the opposite direction according to the phenotype. **d** Venn diagrams indicate poor overlap of gene expression changes when group comparisons were performed with the largest number of genes associated to RES vs CTRL animals. Most significant biological pathways for each group comparison are displayed on the right according to the group comparison color. **e** SS vs RES volcano plot highlights some of the most up- and downregulated genes. Data represent mean ± s.e.m; the number of animals or subjects ($n$) is indicated on graphs. One-way ANOVA followed by Bonferroni's multiple comparison test for behaviours. *$p < 0.05$; **$p < 0.01$; ****$p < 0.0001$. Source data are provided as a Source Data file.

specifically in women[46]. On the other hand, carnitine acyl carnine translocase (*Slc25a20*) is decreased in the PFC of SS females when compared to RES, which is consistent with previous studies reporting decreased acetyl-l-carnitine in MDD patients[47].

Interestingly, both sexes show significant endothelium gene expression changes in the omega-3/omega-6 fatty acid synthesis pathways in the PFC following chronic social stress. However, these alterations were observed in RES vs CTRL for females and between SS vs CTRL in males. This pathway has been extensively studied in human depression as well as in animal models of mood disorders with omega-3 deficiency linked to neuronal atrophy in the medial PFC of male mice, concomitant with anxiety- and depressive-like behaviours[48]. Fatty acids play an important role in the regulation of systemic[49] and endothelium inflammation[50], thus providing an intriguing association between depression-related disruptions in the neuroimmune axis[51] and BBB hyperpermeability. Similarities between the PFC female resilient and male susceptible endothelial transcriptome profiles for this pathway and the fact that both subgroups maintain BBB integrity in this brain region despite chronic stress exposure suggest that it could play a protective role in stress-induced BBB permeability loss. With previous reports linking omega-3 fatty acid consumption with resilience to stress in rodents[52] and humans[53], further studies are warranted to confirm their potential beneficial effect on the neurovasculature and gain mechanistic insights.

A possible mechanism for this sex- and region-specific vulnerability of the BBB is the presence of estrogen receptors on the neurovasculature. Endothelial cells express low levels of functional estrogen receptors[54] and estrogen-coupled receptors can enter the cell nucleus and bind to estrogen-responsive elements (ERE) on specific DNA sequences[55]. Interestingly, ERE and stimulating protein 1 (Sp1) transcription factors were identified on the mouse *Cldn5* gene promoter, which allows estrogen receptors to modulate *Cldn5* transcription through cooperative interactions of Er/Sp1 with ERE/Sp1 elements[56]. While high levels of estrogen render female rats more sensitive to stress-induced PFC dysfunction[57], this was found to be protective in the striatum[58], providing mechanisms to explore for future studies. Our group has recently shown that permissive epigenetic regulation of *Cldn5* expression paired with low endothelium Cldn5-repressive transcription factor forkhead box protein O1 is associated with stress resilience in the NAc of male mice, while increased histone deacetylase 1 level and activity is a mediator of stress susceptibility[8]. Thus, investigating sex-specific epigenetic vascular mechanisms underlying susceptibility vs resilience to stress and BBB hyperpermeability will be important in the future.

Additionally, estrogen-activated receptors have been shown to inhibit the proinflammatory transcription factor NFkB (nuclear factor kappa light chain enhancer of activated B cells), a known regulator of ICAM-1 and E-selectin[59]. Increased circulating proinflammatory cytokines and NFkB were reported in adolescent MDD and bipolar disorder, correlated with depressive symptoms severity[60], and we have identified the proinflammatory

NFkB pathway as a mediator of stress susceptibility in NAc endothelial cells of male[8] but not female mice (Figs. 4 and 5). Accordingly, we observed a sex-specific increase in circulating sICAM-1 levels only in SS male mice vs pre-CSDS at baseline (Supplementary Fig. 10) and *Icam1* levels are increased in the NAc of SS male mice[8] but seem to be reduced in the PFC of SS female mice (Supplementary Fig. 10), reinforcing the idea of sex-specific regulatory mechanisms of BBB integrity, possibly through estrogen-mediated pathways. Social defeat stress increases the expression of adhesion molecules in the male mouse brain, including *Icam-1* and *Sele* the gene encoding for E-selectin[61]. We did observe a trend, that did not reach significance, for higher expression of *Sele* in the PFC of stressed mice (Supplementary Fig. 10) suggesting that the increase in circulating sE-selectin measured in the blood serum of SS females may come, at least partly, from other brain regions or non-CNS sources. Moreover, elevated blood sE-selectin level was reported in the elderly with mild cognitive impairment and depressive mood[62]; however, these studies did not address sex differences. Assessment of BBB leakage using MRI scans in patients suffering from bipolar disorder allowed identification of a subpopulation of patients characterized by worse symptoms including the severity of depression, anxiety, chronicity of illness and decreased global functioning[63]. Despite being commonly used worldwide, it would be unrealistic to apply BBB imaging to a large population scale or in a preventive context highlighting the importance in discovering biomarkers of psychiatric diseases as we aimed to do here. It could be particularly relevant for conditions involving exacerbated inflammation and/or vascular dysfunction and for which MDD prevalence is higher than the general population; for example, stroke or Alzheimer's disease[1].

Many unanswered questions persist regarding BBB adaptations in both health and disease[18]. Our multidisciplinary approach allowed us to identify sex-specific circulating vascular potential biomarkers as well as candidate genes and pathways that could be relevant to inform on MDD diagnosis and develop treatments. Targeting and regulating tight junction protein integrity at the BBB level could represent an innovative strategy to treat mood disorders[43]. However, thinking beyond endothelial cells will be important to better understand the complex biology underlying BBB hyperpermeability in MDD. Single-cell sequencing of *postmortem* brain tissue from individuals with MDD shows important dysfunction in the PFC pyramidal neurons and oligodendrocyte-lineage cells[64], but to our knowledge, this had never been investigated for endothelial cells, smooth muscle cells or pericytes. By characterizing sex- and region-specific neurovascular alterations underlying stress susceptibility in mice and human depression we provide valuable clues and highlight the need to consider sex as a biological variable while defining the role of brain barriers in psychiatric diseases. These findings are also important in the context of cardiovascular diseases (CVD), with evidence supporting a bidirectional relationship between depression and CVD, where depression is a predictor for the

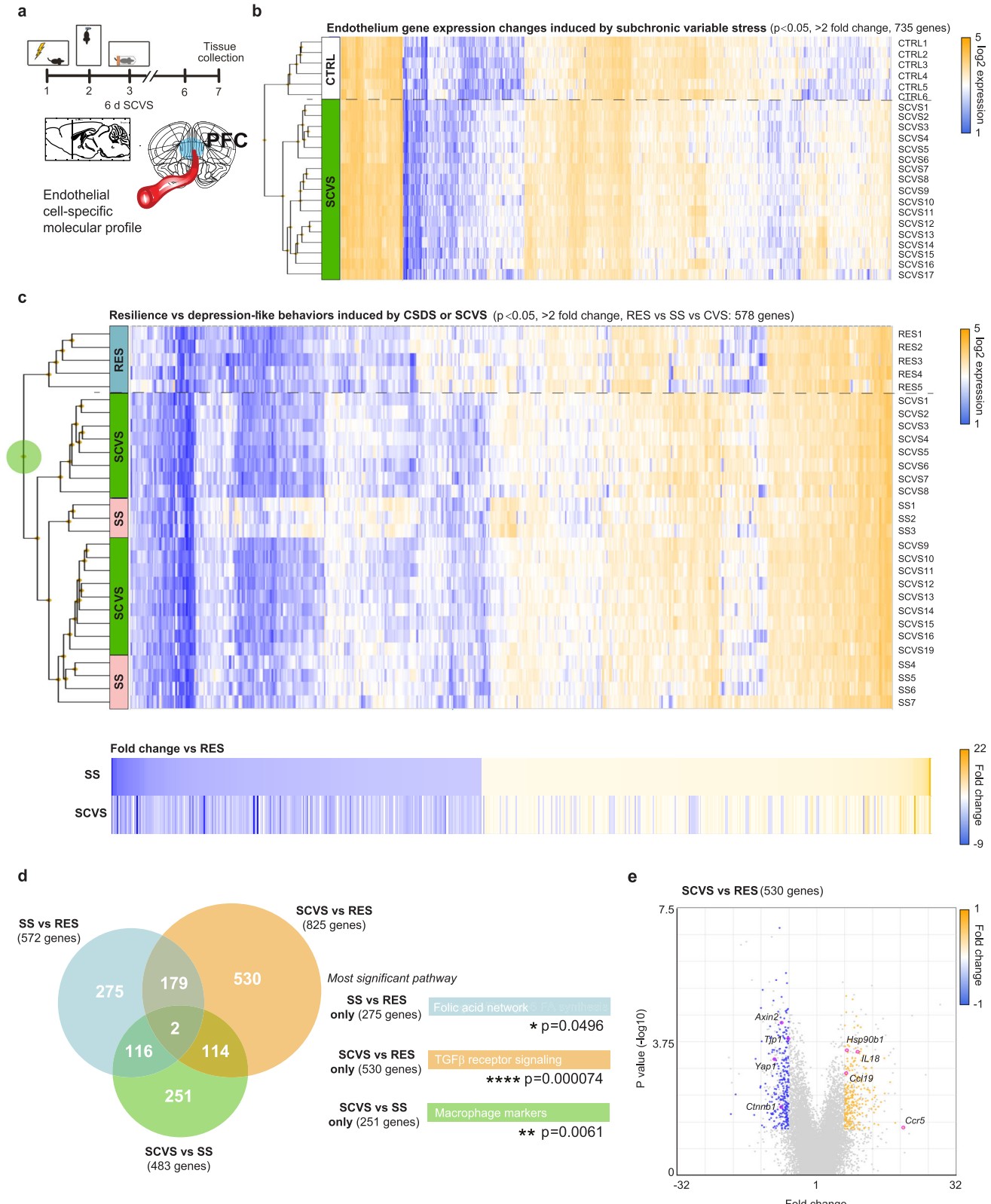

development of CVD, and vice versa[65]. Like in MDD, sex differences in age of onset, symptomatology and treatment response exist in CVD, highlighting the possibility of common etiological basis. Thus, investigating the candidate genes and circulating vascular potential biomarkers associated with depression we report here in populations of CVD patients would be highly relevant and could provide clues into shared underlying

mechanisms of CVD and MDD pathologies. As an example, we observed an increase in sPAI-1, a prothrombotic plasma protein secreted by endothelial tissue, following social and subchronic variable stress in female, but not male, mice (Fig. 6c, d, Supplementary Fig. 10). Dysfunction of this pathway has been proposed as a link between MDD and CVD[66] but underlying mechanisms remain to be elucidated. Genetic variants of the PAI-1 gene

**Fig. 5 Subchronic variable stress induces transcriptome-wide changes in the female PFC endothelium. a** Experimental timeline of the 6-d subchronic variable stress (SCVS) and tissue collection for endothelial cell-specific molecular profiling. **b** Hierarchical clustering heatmap of unstressed controls (CTRL) vs mice subjected to SCVS indicates distinct transcriptomic gene expression (significance was set at ±2-fold change and *p* < 0.05). **c** Hierarchical clustering heatmap of all stressed mice so animals exposed to either 10-d CSDS or 6-d SCVS highlights a level of commonality in gene expression changes between females characterized by depression-like behaviours namely the stress-susceptible (SS) and SCVS group when compared to resilient (RES) subjects (green circle). SS and SCVS females cluster together and normalization of gene expression on RES animals reveals an overlap in the directionality of endothelium transcriptome changes. **d** Venn diagrams show a poor overlap of gene expression changes when group comparisons were performed, particularly for the SCVS vs RES subjects. Most significant biological pathways for each group comparison are indicated on the right. **e** SCVS vs RES volcano plot highlights some of the most up- and downregulated genes. Data represent mean ± s.e.m; the number of animals or subjects (*n*) is indicated on graphs. One-way ANOVA followed by Bonferroni's multiple comparison test for behaviours. \**p* < 0.05; \*\**p* < 0.01; \*\*\*\**p* < 0.0001. Source data are provided as a Source Data file.

(*SERPINE1*) have been associated with MDD[67]. An elevated level of PAI-1 proteins is observed in the serum of individuals with major depression disorder[68], but is also linked with increased risk of ischemic cardiovascular events such as thrombosis and atherosclerosis[69]. Antidepressant treatment decreases PAI-1 expression in the brain of male rats[67]. However, discrepancies exist with PAI-1 deficiency predisposing male mice to depression-like behaviours and resistance to commonly prescribed selective serotonin reuptake inhibitors[70]. Our findings suggest that PAI-1 may be even more relevant for female stress responses, and it was hypothesized that it could play a role in perinatal depression[71], raising interest for future studies on this target to gain mechanistic insights particularly for female rodents and women with MDD.

To sum up, our study shows that chronic stress can induce BBB alterations in the female PFC promoting anxiety, depression-like behaviours, and BBB leakiness (Supplementary Fig. 11). Importantly, loss of tight junction CLDN5 expression was confirmed in *postmortem* brains samples from women with major depressive disorder supporting relevance for human MDD. These alterations were associated with changes in circulating vascular potential biomarkers in the mouse and human blood serum that will have to be explored in future clinical studies and larger human cohorts to confirm translational value. Conversely, despite chronic stress exposure, BBB integrity is maintained in the PFC of RES animals and this could be related to transcriptomic adaptations and activation of protective signalling pathways in the endothelium (Supplementary Fig. 11). Although this project explored neurovasculature-related changes in two brain regions, many other areas involved in emotion regulation remain unexplored and we hypothesize that stress-induced BBB changes go beyond the NAc and PFC. It will also be intriguing to investigate if age-related neurovascular changes such as BBB breakdown, which has been linked to human cognitive dysfunction[72], play a causal role in late-life depression[73] which is more prevalent in women[74].

## Methods

**Mice.** Female and male C57BL/6 (∼20 g) mice were purchased at 7 weeks of age from Charles River and allowed 1 week of acclimation at the housing facility of CERVO Brain Research Center. Sexually experienced retired male CD-1 breeders (∼40 g) of at least 4 months of age (Charles River) were used as aggressors (AGG). All mice were singly housed following CSDS or group-housed during subchronic variable stress and maintained on a 12 h/12 h light/dark cycle throughout. Room temperature was maintained between 19 and 23 °C and humidity was kept around 40–45%. Mice were provided with *ad libitum* access to water and food. All mouse procedures were performed in accordance with the Canadian Council on Animal Care (1993) as well as Université Laval animal care committee (*Certificate #2019-254, VRR-18-052*).

**Estrous cycle identification.** The estrous cycle stage was determined as previously described[75]. Briefly, vaginal lavage was performed with 20 μL of saline and smeared on a slide. The cycle stage was determined by visual inspection of cells under a bright-field microscope (Carl Zeiss).

**Urine collection.** CD-1 mice were placed in metabolic cages (Life Science Equipment) during the dark phase of the light/dark cycle. Urine was collected the following morning, filtered, aliquoted in 0.5 mL tubes and stored at −80 °C until use.

**Chronic social defeat stress (CSDS).** The male 10-day standardized CSDS paradigm was performed as detailed in Golden et al.[14] and our recent studies[7, 8]. For females, CSDS using urine was performed as previously described by Harris et al.[19]. CD-1 mice were screened for aggressive behaviour during inter-male social interactions for three consecutive days based on previously described criteria[14] and housed in the social defeat cage (26.7 cm width × 48.3 cm depth × 15.2 cm height, Allentown Inc) 24 h before the start of defeats (day 0) on one side of a clear perforated Plexiglas divider (0.6 cm × 45.7 cm × 15.2 cm, Nationwide Plastics). Each female mouse was paired with the urine of a particular male CD-1 mouse throughout the entire course of CSDS[19]. Each day, urine was applied to the base of the tail (20 μL), vaginal orifice (20 μL) and upper back (20 μL) of the female mouse then it was immediately subjected to physical interactions with an unfamiliar CD-1 AGG for 10 min. After antagonistic interactions, experimental mice were removed and housed on the opposite side of the social defeat cage divider, allowing sensory contact, over the subsequent 24 h period. Throughout the sessions, mice were monitored for aggressive interactions and mounting behaviours. A session was immediately stopped if persistent mounting or fighting causing physical wounding occurred. Unstressed control mice were housed two per cage on either side of a perforated divider and rotated daily in a similar manner without being exposed to the CD-1 AGG mice. Experimental and control mice were singly housed after the last bout of physical interaction and the social interaction (SI) test was conducted 24 h later. Physical wounding was scored at the time of tissue collection, 24 h after the SI test, and consisted of counting the number of tail bites on the experimental animals as well as the surface area (cm²) of lower back lacerations, if applicable.

As for the 2nd CSDS developed by Takahashi et al.[24] female defeat was performed as previously described. Briefly, a Cre-dependent DIO-Gq-DREADD-expressing AAV was bilaterally injected into the ventromedial hypothalamus of male estrogen receptor alpha (ERα)-Cre-dependent mice. After a recovery period of two weeks, male mice were injected intraperitoneally (i.p.) with clozapine N-oxide (CNO). After 30 min, the female mouse was introduced into the home cage of the male aggressor for 5 min. During the 10 days of defeat, female mice were housed in littermate pairs. Twenty-four hours after defeat, female mice were subjected to a SI test as described below, with a non-injected novel ERα-Cre mouse as the target. All mouse procedures for this model were performed in accordance with the National Institutes of Health Guide for Care and Use of Laboratory Animals and the Icahn School of Medicine at Mount Sinai Animal Care and Use Committee.

**Microdefeat stress and acute defeat stress.** A subthreshold variation of the CSDS protocol was used to evaluate increased susceptibility to stress[7, 76]. Urine from an unfamiliar male CD-1 was applied to experimental C57Bl/6 mice before being exposed to physical interactions with a new male CD-1 AGG for three consecutive bouts of 10 min, with a 15 min rest period between each bout. The SI test was conducted 24 h later. For acute defeat stress, only 1 bout of aggression was performed.

**Social interaction/avoidance test (SI).** SI testing was performed as previously described under red-light conditions[7, 8, 14]. First, mice were placed in a Plexiglas open-field arena (42 cm × 42 cm × 42 cm, Nationwide Plastics) with a small wire animal cage placed at one end. Movements were monitored and recorded automatically for 2.5 min with a tracking system (AnyMaze™ 6.1, Stoelting Co) to determine baseline exploratory behaviour and locomotion in the absence of a social target (AGG). At the end of the 2.5 min, the mouse was removed, and the arena was cleaned. Next, exploratory behaviour in the presence of a novel male CD-1 social target inside the small wire animal cage was measured for 2.5 min and time spent in the interaction and corner zones and overall locomotion were compared. SI ratio was calculated by dividing the time spent in the interaction zone when the

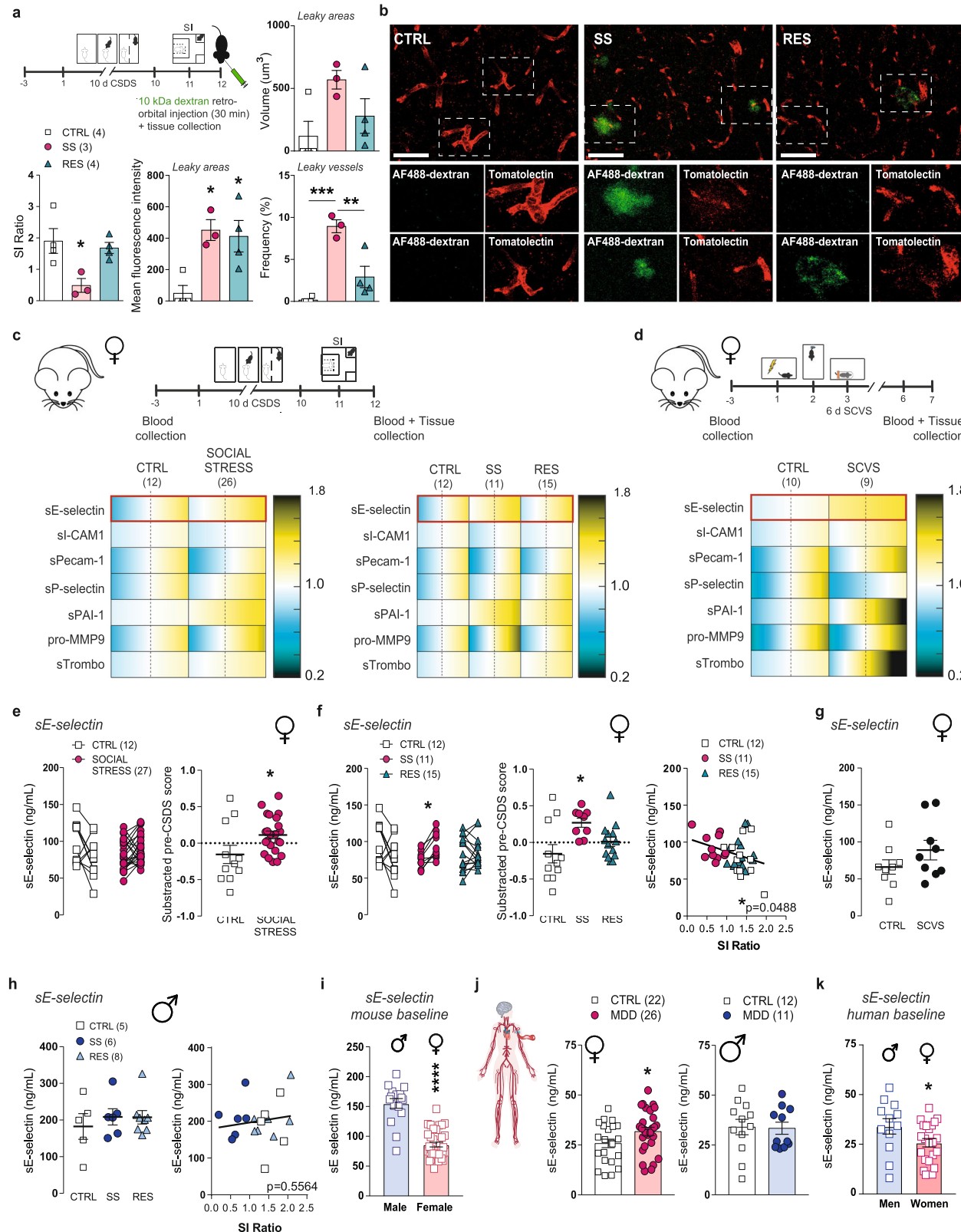

AGG was present vs absent. All mice with a SI ratio below 1.0 were classified as stress-susceptible (SS) and all mice with a SI ratio above 1.0 were classified as resilient (RES).

a tail suspension stress for 1 h (day 2 and 5) and restraint stress, where the animals are placed inside a perforated 50 ml falcon tube, for 1 h within the home cage (day 3 and 6). Tissues were collected 24 h after the last stressor.

**Subchronic variable stress (SCVS).** SCVS was performed as previously described[15, 77], which consisted of three different stressors over 6-days, alternated to prevent habituation. Stressors were administered in the following order: 100 random mild foot shocks at 0.45 mA for 1 h (10 mice into a chamber; day 1 and 4),

**Splash test.** The splash test was used to compare motivated grooming behaviour and performed under red-light conditions as previously described[7, 15]. A 10% sucrose solution was sprayed 3-times on the lower back of the mice and time spent

**Fig. 6 Chronic stress induces BBB leakiness in the female PFC and release of vascular biomarkers in the blood of mice also observed in individuals with MDD. a** Experimental timeline of 10-d chronic social defeat stress (CSDS), social interaction (SI) test and retro-orbital injection of fluorescent-labelled dextran (10 kDa). Following behavioural phenotyping (SI ratio, $*p = 0.0273$), assessment of blood-brain barrier (BBB) permeability with dextran Alexa Fluor-488 dye revealed significant BBB leakiness in the prefrontal cortex (PFC) of stress-susceptible (SS) female mice (fluorescence, $*p = 0.0144$; frequency, $***p = 0.0005$). Scale bars, 20 μm (**b**). **c, d** Experimental timeline of CSDS and subchronic variable stress (SCVS) paradigms and blood collection for multiplex assays. **e, f** Cardiovascular multiplex assays reveals significant upregulation of circulating soluble E-selectin (sE-selectin) when compared to baseline (pre-CSDS) serum levels ($*p = 0.0267$ for SOCIAL STRESS) in SS, but not RES mice ($*p = 0.0131$), and post-CSDS levels correlated with social avoidance (**f**). **g, h** Circulating sE-selectin levels were increased in female mice following 6-d SCVS without reaching significance and remained unchanged in male mice following 10-d CSDS, and no correlation was found with SI ratio. **i**, Baseline serum levels of sE-selectin was ~40% lower in female vs male mice ($****p < 0.0001$). **j, k** Circulating sE-selectin levels were significantly upregulated in women ($*p = 0.0494$) but not men with major depressive disorder (MDD), and levels in healthy controls were ~25% lower in women when compared to men ($*p = 0.0499$) (**k**). Data represent mean ± s.e.m; the number of animals or subjects (*n*) is indicated on graphs. Correlations were evaluated with Pearson's correlation coefficient; 2-group comparisons were evaluated with unpaired t-tests and one-way ANOVA followed by Bonferroni's multiple comparison test for other graphs. $*p < 0.05$; $**p < 0.01$; $***p < 0.001$; $****p < 0.0001$. Source data are provided as a Source Data file.

grooming over 5 min was videotaped and then recorded with a stopwatch by a blinded observer.

**Sucrose preference test**. Anhedonic responses were evaluated with a standard sucrose preference test. Water bottles from home cages were removed and replaced with two 50 mL conical tubes with sipper tops filled with water for a 24 h habituation period. Next, water from one of the 50 mL conical tubes was replaced with a 1% sucrose solution. All tubes were weighed, and mice were allowed to drink *ad libitum* for a 24 h period. Tubes were then reweighed and switched for another 24 h period of *ad libitum* drinking to prevent place preference. At the end of the 48 h testing period, sucrose preference was calculated by dividing the total amount of sucrose consumed by the total amount of fluid consumed over the 2 d of sucrose availability. For sucrose preference in sham and virus-injected mice, each bottle contained doxycycline treatment (2 mg/mL).

**Forced swim test (FST)**. Forced swim test was used to evaluate helplessness as previously described[7, 15]. Mice were placed into a 4 L glass beaker filled with 3 L of water at 25 °C under bright light conditions and videotaped for 6 min. Time spent immobile was measured with a stopwatch by a blinded observer. Immobility was defined as no movement at all or only minor movements necessary to keep the nose above the water versus mobility, which was defined by swimming and struggling behaviours.

**Elevated plus maze**. Mice were placed in a black Plexiglas cross-shaped elevated plus maze (arms of 12 cm width × 50 cm length) under white light conditions for 5 min. The maze consists of a center area, two open arms without walls and two closed arms with 40 cm high walls set on a pedestal 1 m above floor level. Locomotion was monitored and tracked using an automated system (AnyMaze™ 6.1 Stoelting Co). Cumulative time spent in open arms, center, and closed arms as well as total locomotion was compared between groups.

**Transcriptional profiling of mouse tissue**. Nucleus accumbens (NAc) or prefrontal cortex (PFC) samples were collected and processed as described previously[7, 76]. Bilateral 2.0 mm punches were collected from 1-mm coronal slices on wet ice after rapid decapitation and immediately placed on dry ice and stored at –80 °C until use. RNA was isolated using TRIzol (Invitrogen) homogenization and chloroform layer separation. The clear RNA layer was processed using the Pure Link RNA mini kit (Life Technologies) and analyzed with NanoDrop (Thermo Fisher Scientific). RNA was reverse transcribed to cDNA with Maxima-H-minus cDNA synthesis kit (Fisher Scientific) and diluted to 500 μL. For each qPCR reaction, 3 μL of sample cDNA, 5 μL of Power up SYBR green (Fisher Scientific), 1 μL PrimeTime® qPCR primer (Integrated DNA Technologies) and 1 μL ddH20 was added to each well. Samples were heated to 95 °C for 2 min, followed by 40 cycles of 95 °C for 15 s, 60 °C for 33 s and 72 °C for 33 s. Analysis was done using the $\Delta\Delta C_t$ method and samples were normalized to the *Gapdh* mouse housekeeping gene. Primer pairs (Integrated DNA Technologies) are listed in Supplementary Table 1.

**Transcriptional profiling of human tissue**. RNA was isolated using QIAzol Lysis Reagent (Qiagen) homogenization and chloroform layer separation. The clear RNA layer was processed using RNeasy Lipid Tissue Mini Kit (Qiagen) and analyzed with NanoDrop (Biotek). RNA was reverse transcribed to cDNA with iScript Reverse Transcription Supermix kit (Bio-Rad) and diluted to 500 μL. For each qPCR reaction, 3 μL of sample cDNA, 5 μL of Power up SYBR green (Fisher Scientific), 1 μL PrimeTime® qPCR primer (Integrated DNA Technologies) and 1 μL ddH20 was added to each well. Samples were heated to 95 °C for 2 min, followed by 40 cycles of 95 °C for 15 s, 60 °C for 33 s and 72 °C for 33 s. Analysis was done using the $\Delta\Delta C_t$ method and samples were normalized to the *GAPDH*

human housekeeping gene. Primer pairs (Integrated DNA Technologies) are listed in Supplementary Table 2.

**Immunohistochemistry (IHC) and quantification of Cldn5**. Whole brains of mice were frozen using isopentane on dry ice after rapid decapitation. Brains were stored in aluminum foil at −80 °C until use. Brains were embedded in OCT Compound (Thermo Fisher Scientific) and sliced on the cryostat at 20-μm thickness. Slices were post-fixed for 10 min in ice-cold methanol before a quick wash in 0.1 M phosphate-buffered saline (PBS). Sections were then incubated for 2 h in blocking solution, consisting of 4% normal donkey serum (NDS), 1% Bovine Serum Albumin (BSA, GE Life Sciences) and 0.03% Triton X-100 in 0.1 M PBS before overnight incubation with primary antibodies (rabbit anti-Cldn5, 1:250, Life Technologies, #34-1600) in blocking solution. Double immunostaining with CD31 (anti-rat-CD31, 1:100, Invitrogen, #14-0311-85) was performed to allow localization of blood vessels for quantification of tight junction protein levels. After three washes in PBS for 5 min, sections were incubated with anti-rabbit-Cy2 and anti-rat-Cy3 secondary antibodies for 2 h (1:400, Jackson Immunoresearch, #711-225-152, #712-165-153, respectively), washed again three times with PBS. Slices were mounted and coverslipped with ProLong Diamond Antifade Mountant (Invitrogen). One-micrometer-thick z-stack images of the NAc or PFC were acquired on an LSM-880 confocal microscope (Carl Zeiss). Images were taken using a 63× lens with a resolution of 1628 × 1628 and a zoom of 1.0. Pixel size was 0.98 μm in the $x−y−z$ planes, pixel dwell time was 1.98 μs and the line average was set at 2. Maximum Intensity Projection was compared using Image J (NIH) with the region of interest (ROI) defined using CD31 staining.

**Immunohistochemistry (IHC) of Alexa Fluor 488 for BBB leakiness**. Twenty-four hours after the SI test, mice were anesthetized with a mixture of ketamine (100 mg/kg of body weight) and xylazine (10 mg/kg of body weight) and administered 0.1 mL of 1 mg/mL solution of fixable Alexa Fluor 488-dextran (D22910, mW = 10 kDa, ThermoFisher Scientifics) through retro-orbital injection. After 30 min, mice were perfused with ice-cold 0.1 M PBS followed by 4% paraformaldehyde (PFA). Brains were post-fixed overnight in 4% PFA at 4 °C in the dark, then sliced on a vibratome (Leica) at 40 μm thickness. Free-floating sections were washed in 0.1 M PBS and incubated for 2 h in blocking solution (5% NDS in 0.1 M PBS). Sections were then incubated in Lycopersicon Esculentum (Tomato) Lectin DyLight® 594 (1:400, Vector Labs). Sections were washed again three times in PBS, mounted and coverslipped with ProLong Diamond Antifade Mountant (Invitrogen). Thirty-micrometer-thick z-stack images of the PFC were acquired on an LSM-700 confocal microscope (Carl Zeiss). Images were taken using a 10× and 20× lens with a resolution of 1532 × 1532 and 1276x1276x, respectively. Pixel size was 0.42 and 0.25 μm in the $x−y−z$ planes, pixel dwell time was 0.58 and 1.02 μs, respectively, and the line average was set at 1. For analysis of PFC leakiness, five planes from three non-consecutive sections were acquired from each animal. The number and surface area of leaky vessels, intensity of Alexa Fluor 488-dextran leaks and total number of vessels were analyzed with the Imaris 9.6.1 software (Oxford Instruments, UK).

**Human *postmortem* tissue collection**. Human brains were collected and NAc or PFC tissue samples dissected by the Suicide section of the Douglas-Bell Canada Brain Bank (DBCBB; https://douglasbrainbank.ca) under the approval of the institution's Research Ethics Board, and as described previously[7, 8]. All brains are donated to the Suicide section of the DBCBB by familial consent through the Quebec Coroner's Office. In addition to consenting to tissue donation for research and access to relevant (including medical) files, families agree to participate in the psychological autopsy that is conducted two months after the death of their next of kin. Blood toxicology was performed and individuals with evidence of drugs or psychotropic medications were excluded. Individuals with a known history of neurological disorders or head injury were also excluded. Demographic characteristics associated with each sample are listed in Supplementary Table 3. Clinical records and interviews were obtained for

each case and reviewed by three or four mental health professionals to establish independent diagnoses followed by a consensus diagnosis in line with the Diagnostic and Statistical Manual of Mental Disorders (DSM) IV criteria. Groups were matched as closely as possible for gender, age, race, pH, *postmortem* interval, and RNA integrity number[7–9]. All experiments were performed with the approval of Université Laval and CERVO Brain Research Center Ethics Committee *Neurosciences et santé mentale (Project #2019-1540)*.

**Stereotaxic surgery and viral gene transfer**. All surgeries were performed under aseptic conditions using anesthetic as described previously[7, 76]. Mice were anesthetized with a mixture of ketamine (100 mg/kg of body weight) and xylazine (10 mg/kg of body weight) and positioned in a small animal stereotaxic instrument (Harvard Apparatus). The skull surface was exposed, and 30-gauge syringe needles (Hamilton Co.) were used to bilaterally infuse 0.5 μl of virus ($1.0 \times 10^{11}$ infectious unit/mL) expressing either AAV2/9-shRNA or AAV2/9-shRNA-*Cldn5*[7, 27] into the PFC (bregma coordinates: anteroposterior +1.80 mm, mediolateral + /−0.35 mm, dorsoventral –2.35 mm) at a rate of 0.1 μL/min. All mice were allowed to recover for 1 week before a 21-day activation of the viruses with doxycycline treatment (2 mg/mL in drinking water).

**MACS of endothelial cells**. PFC samples were collected following behavioural assessment as described previously[7] or following SCVS. Bilateral 2 mm punches were collected from two adjacent 1 mm coronal slices on wet ice after rapid decapitation and immediately processed for MACS purification[8]. Endothelial cells were enriched from PFC punches by using MACS according to the manufacturer's protocol (Miltenyi Biotec). Briefly, brain punches were dissociated using a neuronal tissue dissociation kit (Miltenyi Biotec, 130-092- 628), applied on a 70 μm MACS smart strainer and washed with HBSS 1×. Thereafter, cells were magnetically labeled with CD45 microbeads (Miltenyi Biotec, 130052301) and passed through a MACS MS column (Miltenyi Biotec, 130-042-201) to proceed to the negative selection of CD45 cells. CD45− fraction was collected and magnetically labeled with CD31 microbeads (Miltenyi Biotec, 139097418) and then passed through the MACS MS column to positively select CD31+ cells. CD45−, CD31+ cells were resuspended in 200 μL of TRIzol for RNA extraction and transcriptome-wide gene-level expression profiling.

**Affymetrix Clariom S transcriptome-wide gene-level expression profiling**. Samples were shipped to Genome Quebec for RNA extraction, quality control with the Bioanalyzer, and gene expression profiling with the Affymetrix Clariom S Pico assay for mice (Thermo Fisher Scientific). Gene expression analysis was performed with the Transcriptome Analysis Console 4.0 provided by Thermo Fisher with the Clariom S assay according to the manufacturer's instructions. To identify significant changes between groups, filters were set at fold change ±2 and $p < 0.05$.

**Flow cytometry of endothelial cells**. Cell fractions obtained following MACS were processed as previously described[8]. Briefly, original, CD45−, CD45+, CD45− CD31−, and CD45− CD31+ cell fraction aliquots were incubated with anti-CD16/32 (BioLegend, 14-0161-82) to block Fc receptors. Cells were then labeled with CD45 APC (BioLegend, 103111) and CD31 PE-CF594 (BD Biosciences, 653616). A viability dye (LIVE/DEAD fixable green, Molecular Probes, L34693) was added to the previous panels to discriminate live cells. Endothelial cells from mouse brain PFC punches were identified as CD45− and CD31+ cells. All analyses were performed on BD LSR II and data were analyzed with FACS Diva software (BD Biosciences, v.6.1.3).

**Blood collection and serum extraction**. Blood samples were collected 72 h before the start of CSDS and SCVS protocols by the submandibular bleeding method. Trunk blood was also collected during tissue sampling. Blood was allowed to clot for at least 1 h before being centrifuged at 10,000 RPM at RT for 2 min. The supernatant was collected and spun again at 3000 RPM for 10 min. The supernatant (serum) was collected, aliquoted and stored at −80 °C until use.

**Milliplex Assays for mouse serum**. MILLIPLEX®96-Well Plate Assays were performed according to the manufacturer's protocol (EMD Millipore). Briefly, the plate was prepped using 200 μL Assay Buffer per well for 10 mins on a plate shaker, at room temperature (RT). Then, 25 μL of each standard and controls were added to the appropriate wells, before adding 25 μL of each sample (diluted 1:20 #MCVD1MAG-77k in Assay Buffer) in duplicates to the rest of the plate, vertically. Then, 25 μL of Antibody-Immobilized Beads was added to each well before being incubated on a plate shaker overnight at 4 °C. The following morning, the plate was washed 3× using a handheld magnet, before adding 25 μL of Detection Antibodies and incubating on a plate shaker for 1 h at RT. Streptavidin-Phycoerythrin was added to each well and incubated once more at RT for 30 min. Finally, the plate was washed 3× and resuspended for 5 min on a plate shaker at RT in 150 μL of Sheath Fluid. The plate was read on a Bio-Plex® 200™ plate reader.

**Human serum sample collection**. All human blood serum samples were provided by Signature Bank from the Centre de recherche de l'Institut universitaire en santé mentale de Montréal (CR-IUSMM) under approval of the institution's Ethics Committee. Samples from volunteers with major depressive disorder were collected at the emergency room of the Institut Universitaire en santé Mentale de Montréal of CIUSSS de l'Est-de-Montreal and samples from healthy volunteers at the CR-IUSMM. All donors provided informed consent and signed a 7-page document detailing the goals of the Signature Bank, participants' involvement (questionnaires and tissue sampling), advantages vs risks, compensation, confidentiality measures, rights as participant and contact information. Subjects with a known history of drug abuse were excluded. Demographic characteristics associated with each sample are listed in Supplementary Table 4. Depressive behaviours were assessed by the Patient Health Questionnaire (PHQ-9), which scores each of the nine Diagnostic and Statistical Manual of Mental Disorders (DSM) IV criteria[78]. All experiments were performed under the approval of Université Laval and CERVO Brain Research Center Ethics Committee *Neurosciences et santé mentale (Project #2019-1540)*.

**ELISA for human serum**. Human soluble E-selectin levels were assayed using quantitative sandwich enzyme immunoassay technique and following the manufacturer's protocol (Quantikine ELISA Human E-selectin/CD62E Immunoassay, DSLE00, R&D Systems). Briefly, assay diluent was added to each well, before adding standards and samples (diluted 1:10) in triplicates and incubating for 2 h at RT. The plate was washed 4 times with wash buffer and Human E-selectin Conjugate was added to each well, before incubating again for 2 h at RT. After four more washes, the Substrate Solution was added and incubated in the dark for 30 min, before stopping the reaction with the Stop Solution. The optical density of the plate was read at 450 nanometers (nm) with wavelength correction at 570 nm on a plate reader. Data was reduced against a four-parameter logistic curve using the Gen 5.0 software and samples with a coefficient of variation above 15% were removed from the analysis. The intra-assay variability of the assay ranged from 5.1 to 6.9% and the inter-assay variability ranged from 7.3 to 8.6%; mean assay sensitivity was 0.009 ng/mL.

**Statistical analysis**. The sample size for CSDS and SCVS mouse cohorts was calculated based on previous studies[7, 8, 15, 76]. All mice were assigned to stress-susceptible (SS) or resilient (RES) groups based on their behavioural profile (no outlier in the study) when compared to unstressed controls (CTRL). SI screening and behavioural tests were performed with automated tracking systems when possible. If not (for splash test, sucrose preference test and forced swim test), scoring was done by experimenters blinded to experimental conditions. Outliers for behavioural testing after SCVS or viral-mediated manipulations—for example, those characterized by impaired locomotion—were identified as being greater than 2 SD from the mean and excluded from statistical analysis. An animal found to be an outlier for 2 or more behavioural tests was completely removed from all statistical analyses. Normality was determined by D'Agostino–Pearson, Shapiro–Wilk and Kolmogorov–Smirnov normality tests using GraphPad Prism software (version 9.0). Most datasets were normally distributed, and then *t*-tests, one-way ANOVAs, two-way ANOVAs, and Pearson's correlations were performed with GraphPad Prism software (version 9.0). Bonferroni was used as a post hoc test when appropriate for one-way and two-way ANOVAs and statistical significance was set at $p < 0.05$. If datasets were not normally distributed a non-parametric Mann−Whitney or Kruskal−Wallis test was used for two or three groups, respectively. Statistical significance was set at $p < 0.05$ with *$p < 0.05$; **$p < 0.01$; ***$p < 0.001$; ****$p < 0.0001$. Values between $p = 0.05$ and $p < 0.10$ were considered as trending without reaching significance. For detailed statistics, please refer to the Excel spreadsheet provided in the Supplementary Information. Visual representation of average and SEM with heat maps was done using Matlab-based software. Individual values were used to compute correlation matrices and $p$ values were determined by Matlab-based software. All quantitative PCR, immunohistochemistry and transcriptional quantification were performed in duplicate in at least two different cohorts of mice.

**Reporting summary**. Further information on research design is available in the Nature Research Reporting Summary linked to this article.

## Data availability
All data supporting the findings of this study are available within the paper and Supplementary Information files. RNA sequencing datasets for male and female mouse PFC endothelium have been deposited in the GEO publicly accessible database under the accession code SuperSeries GSE173823. Raw sequencing data for the male mouse NAc endothelium are available in the Supporting Information of Dudek et al. (2020)[8]. RNA sequencing datasets for human tissue sets have been deposited in the GEO under the accession code GSE102556 as reported in Labonte et al. (2017)[9]. Source data are provided with this paper.

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

## Acknowledgements

The authors thank Valerie Catudal and Melissa Murillo Radtke from Genome Quebec, Isabelle Labonté from the CERVO FACS core, Julie-Christine Lévesque from the CRCHU de Quebec Bioimaging Platform of Infectious Disease Research Center as well as the CERVO Brain Research Centre housing facility staff (special thanks to Louisabelle Gagnon) for their work and support. This research was supported by the Canadian Institutes for Health Research (CIHR, Project Grant #427011 to C.M.), Fonds de recherche du Quebec–Sante (FRQS, Junior 1 Salary Award to C.M.) and C.M. Sentinel North Research Chair funded by Canada First Research Excellence Fund. L.F.P. and F.C. are supported by a National Institute of Health grant R01MH104559. M.C. is supported by grants from the European Research Council (ERC: Retina-Rhythm), The Irish Research Council (IRC), and an SFI Centres grant supported in part by a research grant from SFI under grant number 16/RC/3948 and co-funded under the European Regional Development fund by FutureNeuro industry partners. L.DA., F.N.K., K.A.D. and F.C. are supported by scholarships or fellowships from CIHR, Sentinel North, FRQS, and the Swiss National Science Foundation, respectively. The Douglas-Bell Canada Brain Bank is funded by the RQSHA (FRQS) and a platform support grant from Healthy Brain Healthy Lives (Canada First Research Excellence Fund).

## Author contributions

L.D.A and C.M. designed research; L.D.A, A.C., E.D., F.N.K., K.A.D., B.D., L.F.P., F.C., N.S. and M.L. performed research including behavioural experiments, stereotaxic surgeries, molecular, biochemical, and morphological analysis; N.H. and M.C. provided the AAVs for functional experiments; the Signature Consortium contributed the human blood samples and related demographic data while G.T. and N.M. obtained, characterized and prepared the *postmortem* human samples and related data; L.D.A and C.M. analysed the data and wrote the manuscript which was edited by all authors.

## Competing interests

The authors declare no competing interest.

## Additional information

## Signature Consortium

Frederic Aardema[6], Lahcen Ait Bentaleb[6], Janique Beauchamp[6], Hicham Bendahmane[6], Elise Benoit[6], Lise Bergeron[6], Armando Bertone[6], Natalie Bertrand[6], Felix-Antoine Berube[6], Pierre Blanchet[6], Janick Boissonneault[6], Christine J. Bolduc[6], Jean-Pierre Bonin[6], Francois Borgeat[6], Richard Boyer[6], Chantale Breault[6], Jean-Jacques Breton[6], Catherine Briand[6], Jacques Brodeur[6], Krystele Brule[6], Lyne Brunet[6],

Sylvie Carriere[6], Carine Chartrand[6], Rosemarie Chenard-Soucy[6], Tommy Chevrette[6], Emmanuelle Cloutier[6], Richard Cloutier[6], Hugues Cormier[6], Gilles Cote[6], Joanne Cyr[6], Pierre David[6], Luigi De Benedictis[6], Marie-Claude Delisle[6], Patricia Deschenes[6], Cindy D. Desjardins[6], Gilbert Desmarais[6], Jean-Luc Dubreucq[6], Mimi Dumont[6], Alexandre Dumais[6], Guylaine Ethier[6], Carole Feltrin[6], Amelie Felx[6], Helen Findlay[6], Linda Fortier[6], Denise Fortin[6], Leo Fortin[6], Nathe Francois[6], Valerie Gagne[6], Marie-Pierre Gagnon[6], Marie-Claude Gignac-Hens[6], Charles-Edouard Giguere[6], Roger Godbout[6], Christine Grou[6], Stephane Guay[6], Francois Guillem[6], Najia Hachimi-Idrissi[6], Christophe Herry[6], Sheilah Hodgins[6], Saffron Homayoun[6], Boutheina Jemel[6], Christian Joyal[6], Edouard Kouassi[6], Real Labelle[6], Denis Lafortune[6], Michel Lahaie[6], Souad Lahlafi[6], Pierre Lalonde[6], Pierre Landry[6], Veronique Lapaige[6], Guylaine Larocque[6], Caroline Larue[6], Marc Lavoie[6], Jean-Jacques Leclerc[6], Tania Lecomte[6], Cecile Lecours[6], Louise Leduc[6], Marie-France Lelan[6], Andre Lemieux[6], Alain Lesage[6], Andree Letarte[6], Jean Lepage[6], Alain Levesque[6], Olivier Lipp[6], David Luck[6], Sonia Lupien[6], Felix-Antoine Lusignan[6], Richard Lusignan[6], Andre J. Luyet[6], Alykhanhthi Lynhiavu[6], Jean-Pierre Melun[6], Celine Morin[6], Luc Nicole[6], Francois Noel[6], Louise Normandeau[6], Kieron O'Connor[6], Christine Ouellette[6], Veronique Parent[6], Marie-Helene Parizeau[6], Jean-Francois Pelletier[6], Julie Pelletier[6], Marc Pelletier[6], Pierrich Plusquellec[6], Diane Poirier[6], Stephane Potvin[6], Guylaine Prevost[6], Marie-Josee Prevost[6], Pierre Racicot[6], Marie-France Racine-Gagne[6], Patrice Renaud[6], Nicole Ricard[6], Sylvie Rivet[6], Michel Rolland[6], Marc Sasseville[6], Gabriel Safadi[6], Sandra Smith[6], Nicole Smolla[6], Emmanuel Stip[6], Jakob Teitelbaum[6], Alfred Thibault[6], Lucie Thibault[6], Stephanye Thibault[6], Frederic Thomas[6], Christo Todorov[6], Valerie Tourjman[6], Constantin Tranulis[6], Sonia Trudeau[6], Gilles Trudel[6], Nathalie Vacri[6], Luc Valiquette[6], Claude Vanier[6], Kathe Villeneuve[6], Marie Villeneuve[6], Philippe Vincent[6], Marcel Wolfe[6], Lan Xiong[6] & Angela Zizzi[6]

[6]Institut universitaire en santé mentale de Montréal, Centre intégré universitaire de santé et service sociaux Est, Montreal, QC, Canada.

