## [Peer Review File · Nature Communications]

Reviewers' Comments:

Reviewer #1:

Remarks to the Author:

Manuscript NCOMMS-21-16549-T examines the impact of chronic stress on cerebral vasculature in females. This is an interesting and comprehensive study. Several concerns about analysis and interpretation exist and addressing these would strengthen the report.

1) It is overstated to say that a sex-specific result is identified. This strong language must be removed from the title and results. Only females are used in this study so the title and results should only be framed in this context. The discussion can include comparison to the previous male findings cited throughout, but it is not appropriate to state that this work finds sex-specific changes as only one sex is examined. To make such an assertion in the title and results, both sexes would have to be included and the data compared statistically. It is entirely possible that males would have also been different if run concurrently - this cannot be ruled out. It is also possible that this is a difference in power or effect size and not an actual sex difference. The sex difference is only a qualitative comparison between previously published findings and the existing work. It is not at all accurate to state that these data demonstrate a sex difference. They demonstrate an effect in females, which can be qualitatively compared to previously published male data within the discussion only.

2) The use of the term "sexually dimorphic" is inaccurate. Sex differences in most cases exist on a continuum and true dimorphism is rare. Unless the data from both sexes are in this paper and the values do not overlap between the sexes at all, an interpretation of sexually dimorphic is inaccurate and inappropriate. Further, given that gene expression is in comparison to a control group that is within sex, it is not even possible to compare among papers to determine if there is a possibility of dimorphism or if it is simply a mean difference in a similar range.

3) The use of a male-typical stressor in females with the addition of male urine to the female to induce the behavior should be discussed in the context of limitations and the potential for the reported sex difference to be due to the application of a non-ethologically relevant stressor for one sex but not the other.

4) The potential for the defined resilient behavior to be maladaptive should be discussed.

5) Injury frequency and differences among groups should be discussed especially given the potential for physical injury to cause inflammation which could drive difference in BBB separate from psychological stress.

6) Sample sizes are not clear.

7) The number of outliers dropped and from which groups should be made very explicit and transparent. The use of data subsetting followed but exclusion of experimenter-designated outliers has the potential to create bias in the data set and the utmost transparency in reporting of such analysis paths should be included. It is also important to indicate if outlier removal was conducted before or after data were subsetted.

8) Estrous cycle is observed but it is not apparent how these data were used in the analysis.

9) What was the sex of the stimulus mouse for SI? This essential information and should be factored into the interpretation of the data.

Reviewer #2:

Remarks to the Author:

This is a highly timely and relevant study on the effects of chronic stress (different protocols) on blood brain barrier molecular and morphological alterations in the context of anxiety- and depression-like behaviors, and differentiating alterations in susceptible and resilient individuals. It

focuses in mice and uses tissues from human conditions to verify similitudes of changes observed in the mouse experiments. It represents a tour-de-force study with compelling experimental approaches and sound results.

I only have a couple of issues, which I believe should be addressed before publication:

- The issue of sex-specificity. Although the authors have previously described results from similar experiments performed in male mice, the focus here is -on my view- on female mice. However, the study claims reporting sex-specific effects. (L115: "Sex- and region-specific vascular..."; however, there are no sex-specific data presented here; just data on female mice. L88-89: "This vascular change is sex-specific given that no difference was noted in the PFC of stressed males"; L282: "BBB sex differences"). It is well known that studies performed in different animal facilities and laboratories are frequently not validated in other labs. I believe that this is the case here and, therefore, I would strongly recommend that the study makes claims only regarding the results presented here in females. The comparison with males should be left to the motivation and discussion section, but I do think that it cannot be make part of the conclusions nor title, abstract, etc.
- There is some confusion in the text regarding the use of different human samples/cohort for the replication of different types of observations. Could the authors justify why some studies are done (L135) "in postmortem brain samples from MDD women and men" while other () only in women (L105-107: "we assessed the translational value of our mouse findings by evaluating CLDN5 expression in post-mortem ventromedial PFC tissue from depressed women who died by suicide"). Unless this can be justified, it reads right now as a biased decision.
- Although the manuscript is very well written, appropriateness of citations is sometimes not adequate (L85-86: "...brain region is involved in social behaviors, executive function and decision making – other more relevant papers should be cited here). Please, double check citations for appropriateness and recognition of seminal work.

Reviewer #3:

Remarks to the Author:

This manuscript by Dion-Albert et al reports sex-specific effects on BBB integrity in response to chronic stress that cause anxiety/depressive-like behaviors in mice, also the authors provide validations that similar alterations occur in female human brain in patients diagnosed with major depressive disorder (MDD). The four, major significant findings are: 1) chronic social stress paradigms in female mice cause region specific decrease in expression of BBB TJ protein Claudin5 and barrier disruption in the prefrontal cortex (PFC) (also in nucleus accumbens NAc, though based on their previous studies this happens in male mice). A similar, sex-specific decrease in Claudin5 expression was seen in human PFC with MDD. 2) AAV mediated kd in Cldn5 in PFC of female mice, but not male mice, causes anxiety and depressive behavior. 3) transcriptome profiling of endothelial cells isolated from PFC from control, stressed and resilient female mice ('resilient' mice are exposed to chronic social defeat stress paradigm but do not display anxiety or depressive behaviors) identifies potential causative pathways/genes for BBB breakdown, 4) similar vascular biomarker identified in female mice (sE-selectin) exposed to chronic stress paradigm as human data showing increased sE-selectin blood of females with MDD, this may prove useful in identifying patients at risk for developing MDD.

The data is novel as it shows, for the first time, sex-specific, regional differences in BBB integrity in response to stimuli that causes anxiety/depressive behavior and demonstrates similar sex-specific, regional differences in human brain. Particularly compelling is that kd of Claudin 5 in the PFC of female (but not male) mice illicit depressed/anxiety behavior.

In light of increased reported daily stress and increased diagnosis of MDD in women, identifying cellular and molecular mechanism that may underlie sex differences is important for understanding the pathophysiology of MDD and development of new treatments for a disorder that is challenging to treat. This manuscript would be of interest to broad audiences, including basic and clinical researchers studying behavioral, cellular and molecular underpinnings of MDD and other psychiatric disorders, many of which have sex differences. Further, this would be of interest to

researchers studying the CNS vasculature, in particular regional response to outside stimuli (social stress) and sex differences, which are just beginning to be appreciated in this field.

The manuscript is well written and for the most part the data is rigorous, with appropriate controls and statistical analysis, and convincing. That said there are some moderate and minor critiques, these are detailed below.

Moderate:

- 1) Ln. 107-109, author's mention 'alterations in vessel morphology' however this is difficult to accurately claim without quantitative data looking at many vessels from multiple samples. Please provide quantitative data or remove this statement. Also, similar comment about Ln 134-138, Fig. 2i regarding the morphology of the vessels, I would remove the speculative statement about morphology and 'less vulnerable', there is just not enough data here to support this statement.
- 2) The transcriptome data in Fig. 4 and Fig. 5 reveals paradigm specific gene expression as well as important comparisons between SS and resilient female mice, this is of high interest to understand how the endothelium resilient mice changes to prevent BBB leakage and emergence of anxiety/behavioral phenotypes. A few questions:
 - a. I am a little confused about the data in supplemental Fig. 7 with male mice. It isn't clear what region the gene expression data is from in the male mice (presumably NAc?) also there is no comparison to control endothelium from male and female mice. In other words, does the male SS NAc endothelium have a completely different change in gene expression vs control than female SS vs control in NAc?
 - b. Male SS do not have PFC Claudin5 reduction, neither do resilient females, the latter also do not have leakage (but maybe they have some based on images in Fig. 6b??). Is the male PFC transcriptionally similar to PFC endothelium from resilient females or is the gene expression changes entirely separate? I raise this as this gets at potential underlying mechanisms of sex-differences vs resiliency. In other words, does male endothelium show a 'resiliency' gene expression profile in the PFC or is it not different vs control male mice.
- 3) The BBB integrity assay with 10kDa in Figure 6b (control vs SS vs Res) needs quantification, possibly this can be done on tissue sections from existing samples (ex: blind observer scoring dye extravasation sites per area in different treatment conditions).
- 4) The mannitol experiment is not convincing as presented, it doesn't look any different than control, (Extended Figure 8a,b look exactly the same in provided PDFs as the dextran seems to only be in the vessel in the picture). If they can provide quantitative data of BBB leakage in PFC of SS mice, I would recommend taking this out as it doesn't serve as a convincing positive control and 10kDa dextran is a standard, accepted barrier dye for BBB, can add citations (mannitol BBB opening is typically done through the internal carotid, it's possible that IV injection described doesn't work as well?)
- 5) Fig. 6: E-selectin gene/protein is not normally expressed in healthy brain endothelium, it is upregulated on activated endothelial cells – is the gene for E-selectin (Sele) increase in their transcriptional data? Or is the idea that sE-selectin is coming from a non-CNS, systemic source? I recognize that the cited Sawiki et al paper showed increased in a number of CAMs (including E-selectin, by gene and IHC) however this was in different brain regions and male mice.

Minor:

- 1) 'cldn5' is used throughout the text and figures but this is the gene name for the zebrafish gene, 'Cldn5' should be used for mice (first letter capitalized), please change throughout. Same comment for all mouse genes in Figs 1c, e, 2e, g and throughout the text, first letter should be capitalized.
- 2) Ln 30-31: CVD is brought up in beginning of the discussion but this disease is not part of study and isn't mentioned at all discussion. Consider removing this from the introduction or adding a paragraph to the discussion linking to CVD in the context of the results from their study.
- 3) Fig. 3: AAV-shRNA-Cldn5: From the lab's prior paper performing a similar AAV-KD in male mice on the NAc they showed widespread leakage of small (cadaverine) and large (EB) dyes, increased monocyte recruitment (though not extravasation) and parenchymal IL-6, so protein leakage into the brain. However I was unable to find reference to this data in the manuscript. Referencing their prior work demonstrates the approach (historically) causes functional BBB deficits. It would be best to check this in the female mice with PFC AAV-shRNA-Cldn5 but I would be satisfied with adding this as a line to the results to tell the reader this approach works.
- 4) For the data comparing males and female human samples, it would be helpful to have the male

and female symbol above the data in addition to different colors.

signed: Julie Siegenthaler

Response to referees

We are pleased to submit our revised manuscript now titled "**Blood-brain barrier alterations and vascular biomarkers underlie chronic stress responses in female mice mirrored in human depression**" for consideration as an Article in *Nature Communications*.

We addressed all comments as described in this point-by-point response and thank you, the reviewers, for considering our work interesting, comprehensive, timely, relevant, novel, and compelling. The text was revised according to your recommendations, and we did run additional experiments including quantification of vascular immunostaining in human postmortem brains and blood-brain barrier leakiness in the female mouse prefrontal cortex. The male mouse prefrontal cortex endothelium was sequenced following chronic social stress exposure to directly compare sexes.

The manuscript is now much stronger, and we are grateful for your careful reviews, which have greatly improved our study.

Reviewer #1 (Remarks to the Author):

Manuscript NCOMMS-21-16549-T examines the impact of chronic stress on cerebral vasculature in females. This is an interesting and comprehensive study. Several concerns about analysis and interpretation exist and addressing these would strengthen the report.

1) It is overstated to say that a sex-specific result is identified. This strong language must be removed from the title and results. Only females are used in this study so the title and results should only be framed in this context. The discussion can include comparison to the previous male findings cited throughout, but it is not appropriate to state that this work finds sex-specific changes as only one sex is examined. To make such an assertion in the title and results, both sexes would have to be included and the data compared statistically. It is entirely possible that males would have also been different if run concurrently - this cannot be ruled out. It is also possible that this is a difference in power or effect size and not an actual sex difference. The sex difference is only a qualitative comparison between previously published findings and the existing work. It is not at all accurate to state that these data demonstrate a sex difference. They demonstrate an effect in females, which can be qualitatively compared to previously published male data within the discussion only.

Response: After consideration and discussion we do agree with the reviewer that the report should remain focused on the female findings. Thus, we **modified the title, abstract and results section** to now make comparisons with previous published male data in the discussion only.

cervo.ulaval.ca

2601 chemin de la Canardière, Québec (Québec) G1J 2G3

☎ 418 663-5741

✉ info@cervo.ulaval.ca

Le Centre de recherche Cervo est reconnu par le Fonds de recherche du Québec – Santé

2) *The use of the term "sexually dimorphic" is inaccurate. Sex differences in most cases exist on a continuum and true dimorphism is rare. Unless the data from both sexes are in this paper and the values do not overlap between the sexes at all, an interpretation of sexually dimorphic is inaccurate and inappropriate. Further, given that gene expression is in comparison to a control group that is within sex, it is not even possible to compare among papers to determine if there is a possibility of dimorphism or if it is simply a mean difference in a similar range.*

Response: This is a valid point. We removed all references to a sexual dimorphism associated with depression (**page 1, line 11; page 5, line 113; page 14, line 320; page 15, line 330**).

3) *The use of a male-typical stressor in females with the addition of male urine to the female to induce the behavior should be discussed in the context of limitations and the potential for the reported sex difference to be due to the application of a non-ethologically relevant stressor for one sex but not the other.*

Response: We modified the Discussion to highlight this important limitation (**page 14, line 301-305**).

4) *The potential for the defined resilient behavior to be maladaptive should be discussed.*

Response: This is a valid point. The Discussion was revised, and we now refer to *Wood & Bhatnagar (2015)* which provides an overview of this topic (**page 14, line 309-316**).

5) *Injury frequency and differences among groups should be discussed especially given the potential for physical injury to cause inflammation which could drive difference in BBB separate from psychological stress.*

Response: This is an important point raised by the reviewer. We do control for injury frequency when collecting tissues as now described in the Methods section (**page 21, line 459-461**). Data for female injury frequency after exposure to 10-day chronic social defeat stress were also added in **Extended Figure 1a** and **8b**. No difference was observed between stress-susceptible and resilient females with 33% or less mice per group displaying lower back or tail bites.

6) *Sample sizes are not clear.*

Response: We now provide an Excel spreadsheet detailing statistics, including sample sizes, for all Main and Extended Figures.

7) *The number of outliers dropped and from which groups should be made very explicit and transparent. The use of data subsetting followed but exclusion of experimenter-designated outliers has the potential to create bias in the data set and the utmost transparency in reporting of such analysis paths should be included. It is also important to indicate if outlier removal was conducted before or after data were subsetted.*

Response: We now clearly identify when and why outliers were removed in the detailed Statistics Excel spreadsheet.

8) *Estrous cycle is observed but it is not apparent how these data were used in the analysis.*

Response: Estrous cycle was statistically analyzed as detailed in the Methods using 2-way ANOVA and results presented in the Statistics Excel spreadsheet. No difference was observed except for one female cohort as noted in the Figure Legend (**Extended Fig. 5**).

9) *What was the sex of the stimulus mouse for SI? This essential information and should be factored into the interpretation of the data.*

Response: Female mice were exposed to a male mouse during the social interaction test since the aggression bouts were done by CD-1 males. We added this important information in the Methods (**page 20, line 466**) and do thank the reviewer for highlighting this omission. We now discuss how sex of the stimulus mouse in the social interaction test could affect data interpretation (**page 14, line 305-309**).

Reviewer #2 (Remarks to the Author):

This is a highly timely and relevant study on the effects of chronic stress (different protocols) on blood brain barrier molecular and morphological alterations in the context of anxiety- and depression-like behaviors, and differentiating alterations in susceptible and resilient individuals. It focuses in mice and uses tissues from human conditions to verify similitudes of changes observed in the mouse experiments. It represents a tour-de-force study with compelling experimental approaches and sound results.

I only have a couple of issues, which I believe should be addressed before publication:

- The issue of sex-specificity. Although the authors have previously described results from similar experiments performed in male mice, the focus here is -on my view- on female mice. However, the study claims reporting sex-specific effects. (L115: "Sex- and region-specific vascular..."; however, there are no sex-specific data presented here; just data on female mice. L88-89: "This vascular change is sex-specific given that no difference was noted in the PFC of stressed males?"; L282:

cervo.ulaval.ca

2601 chemin de la Canardière, Québec (Québec) G1J 2G3 ☎ 418 663-5741 ✉ info@cervo.ulaval.ca

Le Centre de recherche Cervo est reconnu par le Fonds de recherche du Québec – Santé

“BBB sex differences”). It is well known that studies performed in different animal facilities and laboratories are frequently not validated in other labs. I believe that this is the case here and, therefore, I would strongly recommend that the study makes claims only regarding the results presented here in females. The comparison with males should be left to the motivation and discussion section, but I do think that it cannot be make part of the conclusions nor title, abstract, etc.

Response: This is a valid criticism from the reviewer, also raised by Reviewer 1. We did **modify the title, abstract and result section** to keep the focus on female findings and now make comparisons to the male published results only in the Discussion.

- There is some confusion in the text regarding the use of different human samples/cohort for the replication of different types of observations. Could the authors justify why some studies are done (L135) “in postmortem brain samples from MDD women and men” while other () only in women (L105-107: “we assessed the translational value of our mouse findings by evaluating *CLDN5* expression in post-mortem ventromedial PFC tissue from depressed women who died by suicide”). Unless this can be justified, it reads right now as a biased decision.

Response: We did clarify that assessment of *CLDN5* expression in post-mortem brain samples were performed for MDD women and men in both brain regions (**page 5, line 107**). About 50% loss was observed for MDD women ($P=0.0068$) in the ventromedial prefrontal cortex vs no difference for men ($P=0.8099$) in this brain region (**page 5, line 107-111**). Conversely, a significant loss of *CLDN5* was observed in the nucleus accumbens for both sexes ($P=0.0275$ and $P=0.0253$ for women and men, respectively) (**page 6, line 134-136**).

- Although the manuscript is very well written, appropriateness of citations is sometimes not adequate (L85-86: “...brain region is involved in social behaviors, executive function and decision making – other more relevant papers should be cited here). Please, double check citations for appropriateness and recognition of seminal work.

Response: We revised the manuscript to include three additional citations related to the prefrontal cortex (PFC) roles (*Anderson et al., 1999; Bechara & Damasio, 2000; Wood & Grafman, 2003*) (**page 4, line 85-86**). Briefly, these papers describe impairment of social and moral behaviors following early damage in the human PFC, decision-making deficits of patients with ventromedial PFC lesions, and an overview of human PFC processing and higher cognitive functions, respectively.

Reviewer #3 - Dr. Julie Siegenthaler (signed) (Remarks to the Author):

This manuscript by Dion-Albert et al reports sex-specific effects on BBB integrity in response to chronic stress that cause anxiety/depressive-like behaviors in mice, also the authors provide

cervo.ulaval.ca

2601 chemin de la Canardière, Québec (Québec) G1J 2G3

☎ 418 663-5741

✉ info@cervo.ulaval.ca

Le Centre de recherche Cervo est reconnu par le Fonds de recherche du Québec – Santé

validations that similar alterations occur in female human brain in patients diagnosed with major depressive disorder (MDD). The four, major significant findings are: 1) chronic social stress paradigms in female mice cause region specific decrease in expression of BBB TJ protein Claudin5 and barrier disruption in the prefrontal cortex (PFC) (also in nucleus accumbens NAc, though based on their previous studies this happens in male mice). A similar, sex-specific decrease in Claudin5 expression was seen in human PFC with MDD. 2) AAV mediated kd in Cldn5 in PFC of female mice, but not male mice, causes anxiety and depressive behavior. 3) transcriptome profiling of endothelial cells isolated from PFC from control, stressed and resilient female mice ('resilient' mice are exposed to chronic social defeat stress paradigm but do not display anxiety or depressive behaviors) identifies potential causative pathways/genes for BBB breakdown, 4) similar vascular biomarker identified in female mice (sE-selectin) exposed to chronic stress paradigm as human data showing increased sE-selectin blood of females with MDD, this may prove useful in identifying patients at risk for developing MDD.

The data is novel as it shows, for the first time, sex-specific, regional differences in BBB integrity in response to stimuli that causes anxiety/depressive behavior and demonstrates similar sex-specific, regional differences in human brain. Particularly compelling is that kd of Claudin 5 in the PFC of female (but not male) mice illicit depressed/anxiety behavior.

In light of increased reported daily stress and increased diagnosis of MDD in women, identifying cellular and molecular mechanism that may underlie sex differences is important for understanding the pathophysiology of MDD and development of new treatments for a disorder that is challenging to treat. This manuscript would be of interest to broad audiences, including basic and clinical researchers studying behavioral, cellular and molecular underpinnings of MDD and other psychiatric disorders, many of which have sex differences. Further, this would be of interest to researchers studying the CNS vasculature, in particular regional response to outside stimuli (social stress) and sex differences, which are just beginning to be appreciated in this field.

The manuscript is well written and for the most part the data is rigorous, with appropriate controls and statistical analysis, and convincing. That said there are some moderate and minor critiques, these are detailed below.

Moderate:

1) Ln. 107-109, author's mention 'alterations in vessel morphology' however this is difficult to accurately claim without quantitative data looking at many vessels from multiple samples. Please provide quantitative data or remove this statement. Also, similar comment about Ln 134-138, Fig. 2i regarding the morphology of the vessels, I would remove the speculative statement about morphology and 'less vulnerable', there is just not enough data here to support this statement.

Response: Additional experiments were performed to provide quantitative data to support our statement as shown on revised Main **Figures 1i** and **2j**.

2) *The transcriptome data in Fig. 4 and Fig. 5 reveals paradigm specific gene expression as well as important comparisons between SS and resilient female mice, this is of high interest to understand how the endothelium resilient mice changes to prevent BBB leakage and emergence of anxiety/behavioral phenotypes. A few questions:*

a. *I am a little confused about the data in supplemental Fig. 7 with male mice. It isn't clear what region the gene expression data is from in the male mice (presumably NAc?) also there is no comparison to control endothelium from male and female mice. In other words, does the male SS NAc endothelium have a completely different change in gene expression vs control than female SS vs control in NAc?*

b. *Male SS do not have PFC Claudin5 reduction, neither do resilient females, the latter also do not have leakage (but maybe they have some based on images in Fig. 6b??). Is the male PFC transcriptionally similar to PFC endothelium from resilient females or is the gene expression changes entirely separate? I raise this as this gets at potential underlying mechanisms of sex-differences vs resiliency. In other words, does male endothelium show a 'resiliency' gene expression profile in the PFC or is it not different vs control male mice.*

Response: These are intriguing and important questions raised by Dr. Siegenthaler. To address them, we performed additional experiments to directly compare endothelium gene expression in the PFC of males and females after exposure to chronic stress. We chose the PFC because the manuscript is now focusing on females as recommended by Reviewers 1 and 2, and this was identified as the most vulnerable brain region in this sex. As shown in revised **Extended Fig.6d-g**, the PFC endothelium transcriptomic changes induced by chronic stress are highly sex-specific with a poor overlap for either SS or RES males and females. Interestingly, the pathway omega-3/omega-6 fatty acid synthesis is the most significant for this brain region in SS males (**Extended Fig.6e**) and RES females (**Fig.4**) possibly in line with low or absence of BBB leakage (now confirmed in revised **Fig.6a** for females and in males in *Menard et al., Nature Neuro, 2017*). We also explored gene expression in unstressed male and female controls as suggested and again noted very different endothelium transcriptomic profiles (**Extended Fig.7d**) confirming the importance to consider sex as a biological variable in vascular-related studies. The main text was revised to integrate these important findings (**page 10, line 216-220; page 15-16, line 344-357**) and we do thank Dr. Siegenthaler for suggesting these additional experiments strengthening the manuscript.

3) *The BBB integrity assay with 10kDa in Figure 6b (control vs SS vs Res) needs quantification, possibly this can be done on tissue sections from existing samples (ex: blind observer scoring dye extravasation sites per area in different treatment conditions).*

Response: We now provide quantification for the BBB integrity assay including average fluorescence intensity, volume, and frequency of leaky areas (**Fig.6a, page 11-12, line 253-256**). Analysis parameters of the Imaris software are detailed in the Methods (**page 27, line 589-592**) and an example provided in **Extended Fig.8c**.

4) The mannitol experiment is not convincing as presented, it doesn't look any different than control, (Extended Figure 8a,b look exactly the same in provided PDFs as the dextran seems to only be in the vessel in the picture). If they can provide quantitative data of BBB leakage in PFC of SS mice, I would recommend taking this out as it doesn't serve as a convincing positive control and 10kDa dextran is a standard, accepted barrier dye for BBB, can add citations (mannitol BBB opening is typically done through the internal carotid, it's possible that IV injection described doesn't work as well?)

Response: As recommended we now provide quantitative data of BBB leakage (**Fig.6a**) and thus, we removed the images related to the mannitol experiment and provide citations to justify the use of a 10 kDa dextran (**page 11, line 243-244**).

5) Fig. 6: E-selectin gene/protein is not normally expressed in healthy brain endothelium, it is upregulated on activated endothelial cells – is the gene for E-selectin (Sele) increase in their transcriptional data? Or is the idea that sE-selectin is coming from a non-CNS, systemic source? I recognize that the cited Sawiki et al paper showed increased in a number of CAMs (including E-selectin, by gene and IHC) however this was in different brain regions and male mice.

Response: This is an intriguing observation and thus, we did run qPCR for this gene on prefrontal cortex samples from our two mouse models of depression and for both sexes (**Extended Fig.10f-g**). Only a trend was observed after social stress in females, with no significant difference after subchronic variable stress, suggesting that sE-selectin may come, at least partly, from other brain regions or a non-CNS systemic source. We now discuss these findings in the revised manuscript (**page 17, line 383-386**).

Minor:

1) 'cldn5' is used throughout the text and figures but this is the gene name for the zebrafish gene, 'Cldn5' should be used for mice (first letter capitalized), please change throughout. Same comment for all mouse genes in Figs 1c, e, 2e, g and throughout the text, first letter should be capitalized.

Response: The manuscript **text, main and extended figures were modified accordingly**.

2) Ln 30-31: CVD is brought up in beginning of the discussion but this disease is not part of study and isn't mentioned at all discussion. Consider removing this from the introduction or adding a paragraph to the discussion linking to CVD in the context of the results from their study.

Response: As suggest, we did add a paragraph in the Discussion linking CVD and MDD in the context of our findings (**page 18-19, line 409-416**).

3) Fig. 3: AAV-shRNA-Cldn5: From the lab's prior paper performing a similar AAV-KD in male mice on the NAc they showed widespread leakage of small (cadaverine) and large (EB) dyes, increased monocyte recruitment (though not extravasation) and parenchymal IL-6, so protein leakage into the brain. However I was unable to find reference to this data in the manuscript. Referencing their prior work demonstrates the approach (historically) causes functional BBB deficits. It would be best to check this in the female mice with PFC AAV-shRNA-Cldn5 but I would be satisfied with adding this as a line to the results to tell the reader this approach works.

Response: We did add a line in the results highlighting that the AAV-shRNA-Cldn5 viruses cause functional blood-brain barrier deficits and leakage of circulating dyes into the brain as recommended (**page 7, line 147-148**).

4) For the data comparing males and female human samples, it would be helpful to have the male and female symbol above the data in addition to different colors.

Response: The male and female symbols were added on **Figure 1, 2 and 6** to improve clarity.

Thank you for your consideration of our revised study.

Reviewers' Comments:

Reviewer #1:

Remarks to the Author:

Dr. Menard and colleagues have been very responsive to the previous reviews and the manuscript conveys important and novel information. I have several remaining minor concerns which are detailed below.

- 1) I believe a word may be missing in the abstract in the sentence that spans lines 14-16.
- 2) The abstract would be more impactful by providing more specific statements related to what is actually shown in this manuscript as opposed to the current broad sentences.
- 3) On line 80, it is implied that males are from this study. The citation of the previous study should be provided and the wording clarified.
- 4) Line 88-89 where it states, "This vascular change is sex-specific..." would be more accurately stated as "may be sex-specific" as this could be a replication bias and is not addressed definitively in the current study. Also, some data in this manuscript show effects in males making the statements appear inconsistent.
- 5) Although it is stated that most data were assessed for normality (line 706), it is not stated which data were not assessed and whether or not data sets were normally distributed. Based on the graphics, several data sets appear to be non-normal distributions in which case the use of mean and sem is not supported and non-parametric tests should be used.
- 6) The alpha is stated as $p < 0.05$ but there are multiple instances where a value above 0.05 is stated to be significant (ex. line 123, 125, 255).
- 7) Sickness behavior could account for the behavioral changes in the cases of injuries and virally altered BBB permeability which could increase entry of LPS to the brain. This possibility could be acknowledged in the discussion. Was body mass stable? Was water and food consumption stable?
- 8) Males are introduced in some of the studies in the manuscript but it is difficult to follow when males are used in the study and when the reference is to previous studies. Also, no males are represented in the methods section other than the aggressors and methods must be added for the males included in these data sets. Also, there are effects in males in some cases so strong statements about sex-specific effects should be re-considered.
- 9) The selection of blood markers was not clearly stated. Why these markers? Also, the discussion does not highlight these findings to the extent that the abstract and title suggest that it might.
- 10) Outliers have not been adequately addressed. It is not clear how many have been removed and from which groups. It is also not clear if the exclusion is made with the sample included in the mean and SD or after removal. The removal of multiple outliers further raises concern about normality of the data and whether or not parametric assessments are appropriate. The median and confidence intervals may be more appropriate.
- 11) The complexity and extensive nature of the studies, while impressive and laudable, does present a challenge for the reader. Perhaps focusing the discussion more on what can be interpreted from these data, as opposed to what future studies are that the lab will investigate, could assist the reader and increase the impact of this important manuscript.

Reviewer #2:

Remarks to the Author:

The authors have addressed all my previous comments in a satisfactory manner. I do not have any further issue.

Reviewer #3:

Remarks to the Author:

This is a revised manuscript by Dion-Albert et al that reports regional (pre-frontal cortex or PFC) BBB integrity loss in response to chronic stress that cause anxiety/depressive-like behaviors in female mice, also the authors provide validation that similar alterations occur in female human brain in patients diagnosed with major depressive disorder (MDD). I was enthusiastic about this work in my review of the prior submission and with new data, analysis and edits addressing my concerns, I am even more excited about the work. The new transcriptional profiling data

comparing Ctl, RES, and SS male and female endothelium from PFC is particularly compelling, it suggests there may be common signaling pathways that underlie 'protection' of the PFC endothelium in response to chronic stress. I am completely satisfied that the authors have addressed all my comments and I congratulate them on compiling this exciting and important study. Reviewed by: Julie Siegenthaler

Response to referees

We are pleased to submit our revised manuscript titled "**Blood-brain barrier alterations and vascular biomarkers underlie chronic stress responses in female mice mirrored in human depression**" for consideration as an Article in *Nature Communications*.

We addressed all additional minor concerns of Reviewer 1 as described in this point-by-point response and would like to thank the three reviewers for their careful assessment of our revised manuscript and figures. The manuscript is now much stronger, and we are grateful for the thorough reviews, which have greatly improved our study.

Reviewer #1 (Remarks to the Author):

Dr. Menard and colleagues have been very responsive to the previous reviews and the manuscript conveys important and novel information. I have several remaining minor concerns which are detailed below.

1) I believe a word may be missing in the abstract in the sentence that spans lines 14-16.

Response: Thank you for noting this omission. The sentence was revised (**page 1, line 14-16**).

2) The abstract would be more impactful by providing more specific statements related to what is actually shown in this manuscript as opposed to the current broad sentences.

Response: This is a valid point. We now specify that viral functional manipulations and endothelium transcriptomic profiling were performed, and BBB leakiness observed, in the prefrontal cortex (PFC) of female mice. Mention of soluble E-selectin as potential biomarker of chronic stress responses and major depression was also added (**page 1, line 15-20**).

3) On line 80, it is implied that males are from this study. The citation of the previous study should be provided and the wording clarified.

Response: The sentence was modified to clarify that we refer to a previous study and the citation was moved from the end to the appropriate part of the sentence (**page 4, line 79**).

4) Line 88-89 where it states, "This vascular change is sex-specific..." would be more accurately stated as "may be sex-specific" as this could be a replication bias and is not addressed definitively in the current study. Also, some data in this manuscript show effects in males making the statements appear inconsistent.

Response: We revised the sentence as recommended (**page 4, line 88**).

5) Although it is stated that most data were assessed for normality (line 706), it is not stated which data were not assessed and whether or not data sets were normally distributed. Based on the graphics, several data sets appear to be non-normal distributions in which case the use of mean and sem is not supported and non-parametric tests should be used.

Response: A column about normality tests was added in the Excel statistics spreadsheet for all Figures and Extended Figures panels. Non-parametric Mann-Whitney or Kruskal-Wallis tests were performed when appropriate as listed in the spreadsheet and described in the Statistical analysis section of the Methods (page 33, line 731-737).

6) The alpha is stated as $p < 0.05$ but there are multiple instances where a value above 0.05 is stated to be significant (ex. line 123, 125, 255).

Response: This is a valid point. We clarified in the Statistical analysis section of the Methods that statistical significance was set at $p < 0.05$ with $*p < 0.05$; $**p < 0.01$; $***p < 0.001$; $****p < 0.0001$ and values between $p = 0.05$ and $p < 0.10$ were considered as trending without reaching significance (page 33, line 738-739). We do agree that the statement about volume of BBB leakiness in the PFC of female SS mice being larger was misleading with a p not reaching significance ($p = 0.0809$) and thus, this part of the sentence was removed (page 12, line 257). Values above $p < 0.05$ were also removed for social interactions and time spent in the open arms after exposure to the subchronic variable stress paradigm in the main text (page 6, line 123-125).

7) Sickness behavior could account for the behavioral changes in the cases of injuries and virally altered BBB permeability which could increase entry of LPS to the brain. This possibility could be acknowledged in the discussion. Was body mass stable? Was water and food consumption stable?

Response: We added this possibility in the Discussion as recommended but highlighted that we did not observe significant differences for wounding (Fig. 1a and 8b) or body weight (data not shown) between stress-susceptible vs resilient mice suggesting that behavioral phenotypes are not due to physical injury (page 14, line 312-318).

8) Males are introduced in some of the studies in the manuscript but it is difficult to follow when males are used in the study and when the reference is to previous studies. Also, no males are represented in the methods section other than the aggressors and methods must be added for the males included in these data sets. Also, there are effects in males in some cases so strong statements about sex-specific effects should be re-considered.

Response: We did clarify that to directly compare stress-induced endothelium transcriptomic changes in the male and female PFC a cohort of male mice was subjected to the 10-day chronic social defeat stress (CSDS) paradigm for the present study. In contrast, male nucleus accumbens endothelium-related gene lists were obtained from our recent study (Dudek et al., 2020) and this is now mentioned in the Results section (**page 10, line 218-222**). We do acknowledge the reviewer concern about our statement of sex-specific effects and thus, modified the sentence accordingly (**page 10, line 215**). Experiments were also performed in male mice for assessment of blood serum biomarkers in the present study and we revised a sentence in the last paragraph of the Results to clarify it (**page 13, line 280**). Finally, we do thank the reviewer for noting that we forgot to add the male CSDS in the Method section and now refer to the standardized protocol detailed in Golden et al., Nature Protocols (2011) and our previous studies (**page 21, line 456 and 473-474**).

9) The selection of blood markers was not clearly stated. Why these markers? Also, the discussion does not highlight these findings to the extent that the abstract and title suggest that it might.

Response: This is an excellent point. We added in the Results section that we took advantage of a commercially available Milliplex assay including 7 analytes related to vascular health (**page 12, line 267-269**). With most of the Discussion centered on the soluble E-selectin findings, this particular potential biomarker was added in the abstract (**page 1, line 20**). We also now discuss the changes observed for plasminogen activator inhibitor 1 (PAI-1) specifically in females and its potential as biomarker for depression with regard to the literature available (**page 19, line 425-438**).

10) Outliers have not been adequately addressed. It is not clear how many have been removed and from which groups. It is also not clear if the exclusion is made with the sample included in the mean and SD or after removal. The removal of multiple outliers further raises concern about normality of the data and whether or not parametric assessments are appropriate. The median and confidence intervals may be more appropriate.

Response: More details have been added in the Excel statistics spreadsheet about the outliers to clarify to which group they were associated. We also highlighted that all mice were assigned to either stress-susceptible or resilient groups following the CSDS paradigm with no outlier removed in the study (**page 32, line 724**). For all behavioral experiments, normality was performed after removal of the outliers according to the mean and SD, then appropriate statistical analyses used with t-test, one-way ANOVA or two-way ANOVA for normally distributed datasets and non-parametric Mann-Whitney or Kruskal-Wallis for others (please refer to the Excel statistics spreadsheet for details as mentioned in the Statistical analysis section of the Methods, **page 32-33, line 727-737**).

11) The complexity and extensive nature of the studies, while impressive and laudable, does present a challenge for the reader. Perhaps focusing the discussion more on what can be

interpreted from these data, as opposed to what future studies are that the lab will investigate, could assist the reader and increase the impact of this important manuscript.

Response: This is a good point. A figure was added to summarize the findings of the current study along with an overview at the end of the Discussion (**Extended Fig.11, page 20, line 439-447**). We hope this will help reach a wider audience and be of broad interest to neuroscientists, psychiatrists and clinicians interested in vascular biology, stress responses and mood disorders but also scientists working in drug development.

Thank you anonymous Reviewer 1 for these additional comments that we believe clarify important points and strengthen the study.

Reviewer #2 (Remarks to the Author):

The authors have addressed all my previous comments in a satisfactory manner. I do not have any further issue.

Response: Thank you anonymous Reviewer 2.

Reviewer #3 - Dr. Julie Siegenthaler (signed) (Remarks to the Author):

This is a revised manuscript by Dion-Albert et al that reports regional (pre-frontal cortex or PFC) BBB integrity loss in response to chronic stress that cause anxiety/depressive-like behaviors in female mice, also the authors provide validation that similar alterations occur in female human brain in patients diagnosed with major depressive disorder (MDD). I was enthusiastic about this work in my review of the prior submission and with new data, analysis and edits addressing my concerns, I am even more excited about the work. The new transcriptional profiling data comparing Ctl, RES, and SS male and female endothelium from PFC is particularly compelling, it suggests there may be common signaling pathways that underlie 'protection' of the PFC endothelium in response to chronic stress. I am completely satisfied that the authors have addressed all my comments and I congratulate them on compiling this exciting and important study.

Response: Thank you Dr. Siegenthaler.

Thank you for your consideration of our revised study.